



# The effect of groundwater depth on topsoil organic matter mineralization during a simulated dry summer in North-West Europe

Astrid Françoys[1,2], Orly Mendoza[1], Junwei Hu[1], Pascal Boeckx[2], Wim Cornelis[1], Stefaan De Neve[1] & Steven Sleutel[1]

[1]Department of Environment, Ghent University, Coupure Links 653, 9000 Ghent, Belgium
[2]Department of Green Chemistry and Technology, Ghent University, Coupure Links 653, 9000 Ghent, Belgium

*Correspondence to*: Astrid Françoys (astrid.francoys@ugent.be)

**Abstract.** With climate change expected to intensify the occurrence and severity of droughts, the control of groundwater table (GWT) depth and capillary rise on topsoil moisture may render a critical driver of biological activity. Consequently, GWT depth could influence topsoil carbon mineralization. In this study, undisturbed 200 cm long soil columns of three different textures (loamy sand, sandy loam and silt loam) were subjected to two artificial GWT depths (–165 cm and –115 cm) in the

laboratory. We examined (1) moisture supply by capillary rise along the soil profile and specifically into the top 20 cm soil, and (2) consequently the effect of GWT on decomposition of an added $^{13}$C-enriched substrate (ryegrass) over a period of ten weeks, with limited water applications representing a dry summer. A 50 cm difference in GWT depth (–165 cm vs. –115 cm) resulted in different topsoil moisture for the sandy loam (31 % vs. 38 % Water-filled pore space (WFPS)) and silt loam (33 % vs. 43 % WFPS) soils. In the loamy sand soil, GWT-induced moisture differences appeared only up to 85 cm above the GWT.

The expected acceleration of mineralization of the added ryegrass under a shallower GWT was not confirmed. In contrast, C mineralization pulses after the wetting events were even higher for the drier –165 cm GWT soils. For the silt loam soil, where capillary rise supply had the largest contribution to topsoil moisture, a lower mineralization rate of the stable $C_{ryegrass}$ pool was also found with shallower GWT. These findings suggest that a potential capillary rise effect of increased topsoil moisture on ryegrass mineralization might have been counteracted by other processes. We postulate that the Birch effect might have been

magnified following the rewetting of drier topsoils under deeper GWT levels, ultimately enhancing mineralization compared to where the soil remains consistently wetter under shallower GWT levels. Based on our findings, including the process of texture-specific capillary supply from the GWT can be required to adequately simulate moisture in the topsoil during droughts as they occurred over the past summers in North-West Europe, depending on GWT and texture combination. However, the net effect on topsoil C mineralization is complex and correct simulation of C mineralization may require further integration of

specific processes connected to fluctuating soil moisture state, such as the Birch effect.



# 1 Introduction

When soil desiccates, soil-water potential becomes strongly negative, and eco-physiological conditions for soil micro-organisms less favorable. In particular intracellular turgor pressure and cellular integrity are no longer guaranteed (Malik and Bouskill, 2022; Wang et al., 2015), while diffusion of substrates and extracellular enzymes becomes impeded (Manzoni et al., 2016). As a result, there is a strong moisture dependency of carbon (C) and nitrogen (N) mineralization in soils. Soil C models therefore simulate moisture content through hydrological modules. As precipitation and irrigation are usually the primary suppliers of topsoil moisture, most models do not account for lateral or upward moisture influxes. However, during prolonged dry periods, drying out of topsoil may lead to establishment of counter-gravity soil suction gradients inducing significant upward redistribution of water from the groundwater table (GWT) to the vadose zone through capillary action, and as such, control topsoil moisture. With progressing climate change throughout Europe, weather patterns are becoming more erratic, with already increased occurrence of unusually lengthy dry periods and even agricultural drought in the Maritime climatic region over the past years (Aalbers et al., 2023; CEU JRC, 2022).

Whether or not moisture supply via capillary rise is a relevant process to be accounted for by soil C models, will not only depend on climate, but also on factors such as the depth of the GWT and soil physical properties. But to date, the effect of the GWT depth and capillary moisture supply has nearly exclusively been studied in relation to crop yields (Awan et al., 2014; Feddes et al., 1988; Kroes et al., 2018; Zipper et al., 2015) and irrigation needs (Babajimopoulos et al., 2007; Jorenush and Sepaskhah, 2003; Prathapar et al., 1992; Yang et al., 2011). For example, Zipper et al. (2015) found optimal maize crop yield at GWT levels of 0.6, 0.8, and 0.9 m depth for sandy loam, loam, and silt loam soils, respectively and attributed this to optimal moisture supply resulting from capillary action. Awan et al. (2014) found that when GWT levels lowered from 100 cm to 150 cm to > 200 cm in silt (clay) loam soils, water supplied by capillary rise to the rootzone of cotton and wheat reduced from 28 % to 23 % to 16 % and 9 % to % 6 to 0 %, respectively. When considering bare soils, simulations of the so-called extinction depth for GWT evaporation resulted in depths of 70, 130 and 420 cm for respectively loamy sand, sandy loam and silt loam soils (Shah et al., 2007). This diverse range of modeling outcomes highlights the site-specific nature of capillary rise, as it not only depends on obvious factors such as soil texture, GWT depth and soil water potential gradients, but also on soil structure and soil profile development. As a result, to the best of our knowledge there exists no robust proof on whether or not, and when, GWT depth might significantly control topsoil heterotrophic activity, which may inform us on the pertinence of accounting for its depth and capillary rise in updated soil C models. To validate simulation results, a few studies have been carried out with parallel small-scale field lysimeter experiments monitoring the soil water balance (Kelleners et al., 2005; Prathapar et al., 1992; Yang et al., 2011). Alternatively, Grünberger et al. (2011) injected a deuterium enriched solution to the GWT to follow capillary rise in arid areas. Both approaches, however, are labor intensive and/or require high investments and technical expertise. Li et al. (2022) instead simply excluded upward capillary moisture transport in a field trial on crop residue decomposition by placing a 5 cm gravel layer at a depth of 50 cm, and found that for sandy soils a GWT depth at just 60 cm was not shallow enough to notably provide the top 25 cm soil with capillary moisture. However, this approach required



disturbance of the topsoil and moreover the artificial break of capillary rise also unintentionally cancelled out unsaturated downward moisture redistribution. Most importantly perhaps, the main impediment of observational field approaches, such as the ones listed above, is their inability to control ambient factors such as GWT depth, precipitation, temperature and relative humidity. This limitation restricts our ability to study the effect of individual components of the soil water balance like capillary rise.

As an alternative, a handful of laboratory-scale experiments have sought to infer the capillary moisture impact on soil biogeochemical processes. Rezanezhad et al. (2014) and Fiola et al. (2020) found that highest C mineralization was found at transient redox conditions above the capillary fringe, where moisture and oxygen are in balance. However, due to the small scale of the used setups (packed soil columns of 45 cm and 30 cm length, respectively) an appreciation of capillary rise was not possible. Malik et al. (1989), Shaw and Smith (1927) and Lane and Washburn (1947) assembled larger packed soil columns

to determine maximum capillary rise height as a function of soil texture. They found capillary moisture supply up to 149 cm (loamy sand soil), 183 cm (sandy loam soil) and 359.2 cm (silt soil), respectively. But as those columns were repacked from sieved soil, soil structure was disrupted and in-field occurring heterogeneity and macropores were not well represented, while neither the impact on C mineralization was assessed (Lewis and Sjöstrom, 2010). In sum, there is no clear empirical evidence of the control of moisture supply by capillary rise on topsoil organic matter (OM) mineralization.

Our aim was to study if, during a (simulated) period of drought, there would be a significant effect of capillary rise from the GWT on topsoil moisture and OM mineralization for loess deposited arable lands in North-West Europe. We designed a setup wherein excavated 200 cm long undisturbed soil columns were incubated in the laboratory with ambient factors being regulated and soil moisture monitored. Columns of three soil textures were subjected to minimal watering events representing a dry summer and two GWT depths to study the interaction between both factors and to provide a representative depiction for our

study region, i.e. North-West Europe. The decomposition of an introduced substrate, i.e. $^{13}C$-enriched ryegrass, was monitored through $CO_2$ headspace measurements. We hypothesized that a deeper GWT would result in reduced topsoil moisture content and as a result, C mineralization in the topsoil would be relatively inhibited compared to the treatment with shallower GWT. We expected an increasing susceptibility to reduced moisture of the C mineralization with coarser soil texture as water losses by evaporation would be less compensated by capillary moisture input. Although physicochemical protection of OM is stronger

in finer-textured soils, we expected that such direct effect of soil texture on mineralization of the added OM would be of less importance in the short term (10 weeks) as opposed to regulation of soil moisture by the soil texture and GWT depth combinations.

## 2 Material and methods

### 2.1 Study area and undisturbed soil column collection

Undisturbed soil columns were collected from three croplands in North-West Belgium with different soil texture, that is a loamy sand soil (50°55'43.8"N 3°32'54.7"E, Kruisem), a sandy loam soil (50°57'47.3"N 3°45'37.2"E, Bottelare) and a silt loam



soil (50°55'15.8"N 3°45'03.0"E, Oosterzele). The fields were chosen based on their GWT depths between 100 and 200 cm in the year prior to our experiment. A cylinder auger set and a motorized percussion hammer (Eijkelkamp, The Netherlands) were employed to excavate the soil columns (Ø: 9.7 cm). First, 100 cm long columns were taken from the soil surface down to –

100 cm depth. Then, a 100 cm deep pit (2 m²) was dug to likewise collect columns from –100 cm to –200 cm depth from withing the pit. Additionally, disturbed topsoil samples (from 0 cm to –20 cm) were collected near each sampling location. The soil columns were carefully transferred into PVC liners (2 half pipes tied together by strong cable ties) and transported to the laboratory. The two soil columns taken within one soil profile were then combined in 200 cm long PVC tubes, that were cut lengthwise beforehand to enable the transfer and later sensor installation. On the inner walls of the PVC tubes a waterproof

foil was applied to avoid moisture losses. To ensure a hydrological connection between the two 100 cm samples, 5 cm of silt clay loam soil (17 % sand, 48 % silt and 35 % clay) was added in between. Additionally, small breaks were restored by using this soil, but when confronted with larger damage, cores were discarded. Therefore, per field and depth increment, six columns were deliberately sampled to only retain the four best structurally intact replicates for our experiment. After closing the PVC tubes, cable ties were tightly applied to attain a solid setup and avoid sidewall effects (Lewis and Sjöstrom, 2010).

**2.2 Experimental design**

**2.2.1 Laboratory setup and incubation**

A soil incubation experiment was set up in which mineralization of a model $^{13}$C-labeled substrate amended to the topsoil was followed as a function of two constant GWT depths, viz. –165 cm and –115 cm (relative to the soil surface) (Fig. 1). The setup was installed in a temperature controlled ($20.8 \pm 0.5$°C) dark room. The GWT treatments were consecutively applied to the

same columns, resulting in two incubation periods under identical laboratory conditions. The top 20 cm soil was removed, such that columns of 180 cm length remained. The soil columns were submerged in 220 L barrels with tap water, to a height of 35 cm or 85 cm height, representing the –165 cm or –115 cm GWT depth, respectively. The soil columns were allowed to stabilize for 27 days. The loamy sand columns were covered by parafilm after 16 days to avoid further drying out. After this stabilization period, the disturbed topsoil samples (0 to –20 cm) collected near each sampling location in the field were mixed

with $^{13}$C-enriched ryegrass and repacked to a 20 cm layer on top of the undisturbed soil columns as described in more detail below.

We have chosen a model OM substrate (*in casu* $^{13}$C-labeled clipped ryegrass) over a comparison of native soil OM mineralization, as the effect of GTW depth, soil texture and their interaction on OM mineralization could also be function of inherently different soil OM quality and quantity among the three studied soils. The extra disturbed top 20 cm soil was first

dried and sieved at 4 mm and visible root fragments were manually removed. Subsequently, the three soils were pre-incubated for one week at 20.8°C at a moisture content of 0.15 , 0.22 and 0.28 m³ m⁻³ for the loamy sand, sandy loam and silt loam columns, respectively. The model substrate, i.e. $^{13}$C-labeled ryegrass ($\delta^{13}$C = +44.93 $\pm$ 1.65 ‰), was added at a dose of 1.5 g C kg⁻¹ ($C_{ryegrass}$ = 38.74 $\pm$ 0.99 %, C:N = 12.8) and mixed thoroughly. The production of this $^{13}$C-enriched ryegrass is described





by Li et al. (2023). To exclude differences in N availability between the various soil texture and GWT depth treatment
combinations, $KNO_3$ (dissolved in water) was added at a dose equivalent to 100 kg N ha[-1]. Each of these texture-specific soil-grass mixtures was then gently packed on top of the columns to bulk densities of 1.50, 1.45 and 1.40 g cm[-3] for the loamy sand, sandy loam and silt loam columns, respectively. Volumetric water sensors (type EC-5, 5TM, Teros10, Teros12, from Decagon and METER group, USA) were installed at different depths (–10, –30, –60, –85 and –120 cm) by puncturing the sensor rods through the waterproof foil. Prior to use, all sensors were calibrated for the three specific soil textures. Dataloggers, type ZL6
(METER group, USA), were used with a log frequency of one hour. Water levels in the barrels were kept constant daily with the help of a float and a time-of-flight sensor (Adafruit, VL53L0X) combined with a Raspberry Pi (small single-board computer). A plastic grid with a mesh size of 5 mm placed between the undisturbed soil column and the repacked topsoil allowed its removal after completion of the first incubation batch while preserving the structure and hydraulic contact with the underneath undisturbed columns. Similarly, for the second GWT treatment, fresh topsoil mixed with ryegrass was once again
added. Rainfall was simulated by gently adding 85 or 128 mL over a 30-minute period every 14 to 21 days, equivalent to a dose of 25 mm month[-1]. With this watering scheme, we simulated a drier than usual (78 mm month[-1]; 30-year Belgian average between the 21[st] of June to 20[th] of September over 1991-2020) local Maritime climate summer, without exceeding the actual measured lowest extreme of 5.2 mm month[-1] only observed in July 2022 (Royal Meteorological Institute, 2022). At the end of each GWT treatment incubation batch, the packed topsoil was removed and its gravimetric moisture content was determined.
Both initial and final gravimetric moisture contents were converted into volumetric water content ($\theta_V$) using the applied bulk densities and compared with the sensor values (Table A1). Deviating measurements were found for three sandy loam columns during the –115 cm GWT treatment, presumably due to air entrapment around the sensor rods after installation. To rectify this discrepancy, a correction was applied using linear regression (Fig. A2).




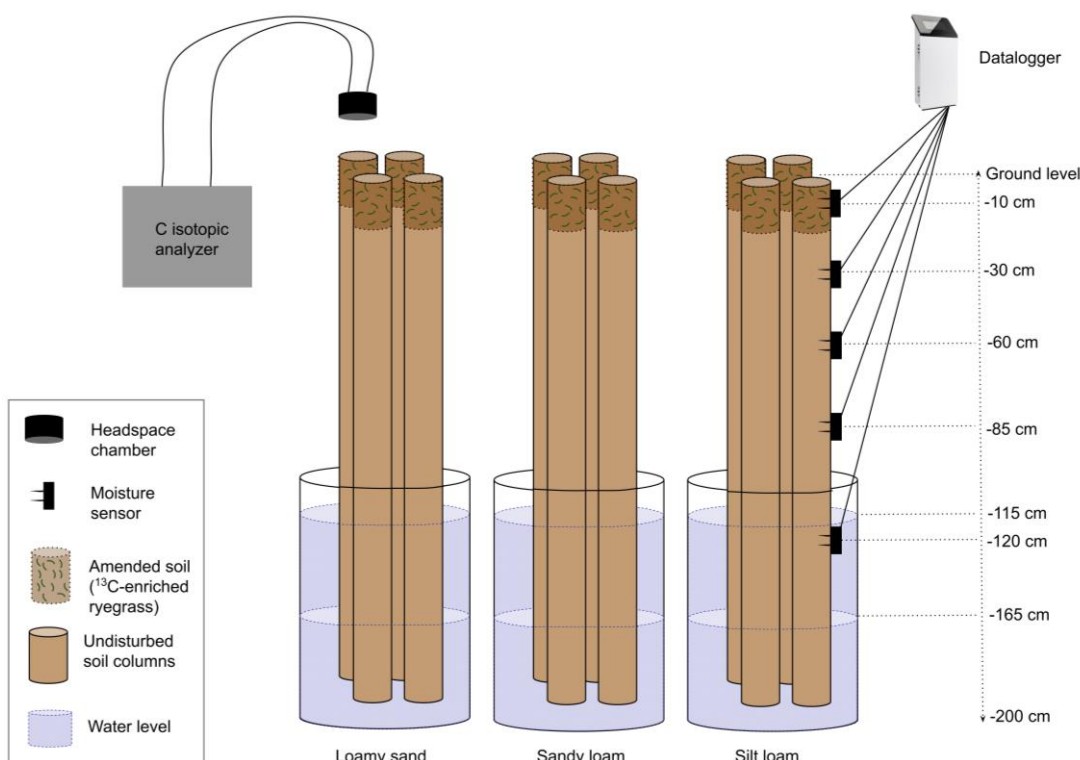


**Figure 1: Schematic overview of the laboratory setup.**

### 2.2.2 Soil CO₂ efflux measurements and calculations

Soil $CO_2$ efflux was regularly measured to record soil C mineralization and infer the progress of the cumulative amount of

$C_{ryegrass}$ mineralized over time. For $CO_2$ efflux measurements, a G2201-i Cavity Ring-Down Spectrometer (CRDS) (Picarro, USA) was consecutively connected via tubing to an air tight lid attached on top of the soil columns. The lid contained two sampling ports (gas in- and outlet), a vent tube (Ø: 1 mm) to minimize air pressure differences and a 9V battery-powered fan on the inside to ensure air circulation inside the headspace between soil surface, PVC tube and lid. The headspace volume varied between 0.52 and 0.82 L and covered an area of 73.90 $cm^2$. The CRDS recorded changes in the headspace $CO_2$

concentration and $\delta^{13}C$ value every 2–3 seconds. $CO_2$ efflux rates were calculated as the slope of the linearly increasing $CO_2$ concentration over a 7 to 15 minutes time interval per soil column and were converted to a mass-based unit (mg $CO_2 \cdot C$ kg soil$^{-1}$ h$^{-1}$) using the ideal gas law. To determine the $\delta^{13}C$ (‰) value of emitted $CO_2$ ($\delta^{13}C \cdot CO_2$, Eq. (1)), the Keeling plot method was applied in which measured $\delta^{13}C$ values are plotted against the inverse $CO_2$ emission concentrations and the $\delta^{13}C$ of emitted $CO_2$ is obtained from the y-axis intercept of a linear model fitted to the data (Keeling, 1958). Measurements were



made on days 1, 2, 3, 5, 7, 9, 11, 13, 15, 20, 22, 24, 27, 29, 34, 36, 38, 42, 45, 48, 52, 57, 59, 62, 66 and 69 of each of both incubation batches. The CRDS was recalibrated at onset of each incubation using two certified standard gases of 400 ppm and 2500 ppm $CO_2$ with $\delta^{13}C$ values of respectively –8.6 ‰ and –51 ‰. The contrasting $\delta^{13}C$ values of the added $^{13}$C-labeled ryegrass (+ 44.93 ± 1.65 ‰) and soil organic carbon (– 25.68 ± 0.23 % for loamy sand, – 25.57 ± 0.17 % for sandy loam, and – 24.79 ± 0.11 ‰ for silt loam soils, Table 1) allowed to partition the total $CO_2$ efflux ($CO_2 \cdot C_{total}$) into a share resulting from

ryegrass decomposition ($CO_2 \cdot C_{ryegrass}$) and from native SOC ($CO_2 \cdot C_{SOC}$) (µg $CO_2$-C kg$^{-1}$ soil) with an isotope mixing model following Eq. (1):

$$CO_2 \cdot C_{ryegrass} = \frac{\delta^{13}C \cdot CO_2 - \delta^{13}C \cdot CO_{2 \cdot SOC}}{\delta^{13}C \cdot CO_{2 \cdot ryegrass} - \delta^{13}C \cdot CO_{2 \cdot SOC}} * CO_2 \cdot C_{total} \tag{1}$$

The isotopic signature of $CO_2$ emitted from either end member, i.e. $\delta^{13}C \cdot CO_{2 \cdot ryegrass}$ and $\delta^{13}C \cdot CO_{2 \cdot SOC}$, were analyzed in

ancillary soil incubations as in Mendoza et al. (2022). The $\delta^{13}C \cdot CO_{2 \cdot SOC}$ was determined in parallel 20 cm packed soil columns with no ryegrass added. For $\delta^{13}C \cdot CO_{2 \cdot ryegrass}$, such soil columns were amended with a high dose of ryegrass (3 g C kg$^{-1}$ soil; indicated as "high"), and the following Eq. (2) was applied:

$$\delta^{13}C \cdot CO_{2 \cdot ryegrass} = \frac{CO_2 \cdot C_{high} * \delta^{13}C \cdot CO_{2\,high} - CO_2 \cdot SOC \cdot C_{SOC} * \delta^{13}C \cdot CO_{2\,SOC}}{CO_2 \cdot C_{high} - CO_2 \cdot C_{SOC}} \tag{2}$$


Emission measurements in these ancillary incubations were made on days 2, 7, 15, 29, and 52.

From ryegrass-derived $CO_2$ efflux ($CO_2 \cdot C_{ryegrass}$ h$^{-1}$) cumulative amounts of mineralized $C_{ryegrass}$ (in µg C kg$^{-1}$ soil) were calculated by integrating these mineralization rates over time intervals defined as half of the period before the previous measurement and half of the period until the next measurement. To describe the kinetics of mineralized ryegrass over time, 

relative to the added amount of ryegrass in % $C_{ryegrass}$, the following parallel first-zero-order kinetic model was used (Sleutel et al., 2005; Zacháry et al., 2018), Eq. (3):

$$Cumulative\ C_{ryegrass}\text{-}min\ (t) = C_f * \left(1 - e^{-k_f * t}\right) + k_s * t \tag{3}$$

where $C_f$ (% of $C_{ryegrass}$) and $k_f$ (day$^{-1}$) are parameters representing the easily mineralizable $C_{ryegrass}$ pool and the first-order mineralization rate, respectively, while $k_s$ (% of $C_{ryegrass}$ day$^{-1}$) is the zero-order mineralization rate constant of a more stable $C_{ryegrass}$ pool.



## 2.3 Column analyses: physicochemical soil properties

At the end of the experiment, the 200 cm tubes were opened carefully by relieving cable ties, taking off one of both PVC half-pipes and opening the waterproof foil. Then, undisturbed soil samples (Ø: 5 cm, h: 5.1 cm) were taken by pressing Kopecky rings in the soil columns near the moisture sensor locations, i.e. at –15 to –10 cm; –35 to –30 cm; –55 to –50 cm; –85 to –80 cm and –125 to –120 cm depth. For each depth layer a soil-water retention curve was determined by measuring water contents of these ring samples on a silica sand tension table and pressure plates (Eijkelkamp, The Netherlands) at different matric

tensions (–1, –3, –7, –10 and –33, –100, –1500 kPa, respectively). Average water retention curves per texture and depth (Fig. B1) were further used to convert measured mean volumetric water contents in matric potentials, expressed as matric head in units of cm water height (cm WH). The latter were used to calculate hydraulic head differences (ΔH) between two adjacent sensor positions above the GWT, with hydraulic head the sum of matric head and gravitational head. They were used as an indicator for the moisture transport direction: positive ΔH values indicate a net upward (capillary) water transport, while

negative ΔH values signify an overall downward water transport. Soil texture, OC content and $pH_{H2O}$ were determined on homogenized subsamples from –20 to –50 cm, –50 to –100 cm, –100 to –150 and –150 to –200 cm layers. Values for $\delta^{13}C$ were only determined for the repacked topsoil. Physicochemical properties of the soil columns are listed in Table 1.

## 2.4 Statistical analyses

Statistical analyses were made with R Studio statistical software, version 4.1.2 (The R foundation, Austria). Reported values

represent means ± standard errors. Linear mixed-effect models (R package "nlme", function "lme") (Pinheiro et al., 2023) in combination with estimated marginal means (R package "emmeans", function "emmeans") (Lenth et al., 2024) were used to detect statistical differences in average moisture content and $C_{ryegrass}$ mineralization rates between GWT treatments per texture over the entire incubation period. In this model, the GWT treatment was set as fixed factor, while column replicates (n = 4) were added as random intercept to represent the grouped structure of the experimental setup and an autocorrelation factor was

included to account for temporal autocorrelation between measurements as well (Schielzeth and Nakagawa, 2013). Diagnostic plots for the linear mixed-effects fit were visually examined (R package "nlme", function "plot.lme"). Additionally, paired two-tailed t-tests were used to compare $C_{ryegrass}$ mineralization rates between GWT treatments per measurement day (after checking assumptions of normality and homoscedasticity). Goodness-of-fit of the parallel first-zero-order kinetic model was assessed through the Nash–Sutcliffe model efficiency coefficient (NSE) (R package "ie2misc", function "vnse") (Embry et

al., 2023). To detect any effect of GWT treatment and texture on the kinetic parameters of the first-zero-order C mineralization model ($C_f$, $k_f$ and $k_s$) and cumulative $C_{ryegrass}$-min at the end of the incubation, once more linear mixed-effect models (R package "lme4", function "lmer") (Bates et al., 2023) were applied in combination with estimated marginal means (R package "emmeans", function "emmeans"). This time, both GWT and texture were set as fixed factors, while the separate columns (n = 12) were set as a random effect to allow for a pairwise comparison. The normality assumption for the residuals was tested

using a simulation based approach (R package "DHARMa") (Hartig, 2020).



**Table 1: Physicochemical properties of the undisturbed soil columns (n = 4).**

| Location cropland | Depth below the surface from; to (cm) | Soil texture [b] | | | USDA soil texture class | OC [c] (g kg$^{-1}$) | $\delta^{13}$C [d] (‰) | pH$_{H2O}$ [e] (-) | Sensor depth (cm) | BD [f] (g cm$^{-3}$) |
|---|---|---|---|---|---|---|---|---|---|---|
| | | Sand (%) | Silt (%) | Clay (%) | | | | | | |
| Kruisem | 0 ; –20 [a] | | | | | 11.2 ± 0.3 | – 25.68 ± 0.23 | | –10 | 1.40 |
| | –20 ; –50 | 86.2 ± 0.3 | 10.4 ± 0.6 | 3.3 ± 0.3 | Loamy sand | 5.0 ± 1.0 | | 7.3 ± 0.0 | –30 | 1.54 ± 0.02 |
| | –50 ; –100 | 89.5 ± 1.6 | 8.4 ± 1.4 | 2.1 ± 0.2 | Sand | 0.6 ± 0.2 | | 7.3 ± 0.0 | –60 | 1.58 ± 0.01 |
| | –100 ; –150 | 85.3 ± 3.2 | 8.1 ± 2.6 | 6.6 ± 1.4 | Loamy sand | 0.2 ± 0.1 | | 7.6 ± 0.0 | –85 –120 | 1.52 ± 0.01 1.66 ± 0.01 |
| | –150 ; –200 | 76.6 ± 5.7 | 14.1 ± 5.4 | 9.3 ± 0.7 | Sandy loam | 0.5 ± 0.1 | | 7.7 ± 0.1 | | |
| Bottelare | 0 ; –20 [a] | | | | | 10.0 ± 0.4 | – 25.57 ± 0.17 | | –10 | 1.45 |
| | –20 ; –50 | 60.6 ± 0.8 | 33.1 ± 0.9 | 6.3 ± 0.3 | Sandy loam | 3.6 ± 0.6 | | 7.4 ± 0.0 | –30 | 1.55 ± 0.01 |
| | –50 ; –100 | 62.8 ± 3.8 | 26.6 ± 3.6 | 10.6 ± 1.8 | Sandy loam | 0.7 ± 0.1 | | 7.6 ± 0.0 | –60 | 1.75 ± 0.02 |
| | –100 ; –150 | 47.7 ± 1.0 | 36.7 ± 1.3 | 15.6 ± 0.4 | Loam | 0.5 ± 0.1 | | 7.5 ± 0.0 | –85 –120 | 1.65 ± 0.02 1.65 ± 0.01 |
| | –150 ; –200 | 66.5 ± 4.6 | 23.0 ± 3.7 | 10.5 ± 0.9 | Sandy loam | 0.2 ± 0.1 | | 7.3 ± 0.0 | | |
| Oosterzele | 0 ; –20 [a] | | | | | 9.8 ± 0.2 | – 24.79 ± 0.11 | | –10 | 1.50 |
| | –20 ; –50 | 12.2 ± 1.2 | 68.8 ± 0.5 | 18.9 ± 0.8 | Silt loam | 4.3 ± 0.6 | | 7.1 ± 0.1 | –30 | 1.61 ± 0.02 |
| | –50 ; –100 | 15.6 ± 3.8 | 66.8 ± 2.9 | 17.6 ± 0.9 | Silt loam | 1.4 ± 0.2 | | 7.3 ± 0.2 | –60 | 1.60 ± 0.02 |
| | –100 ; –150 | 22.5 ± 4.3 | 60.7 ± 3.3 | 16.8 ± 1.1 | Silt loam | 0.9 ± 0.2 | | 7.6 ± 0.2 | –85 –120 | 1.54 ± 0.01 1.67 ± 0.01 |
| | –150 ; –200 | 16.6 ± 2.1 | 66.2 ± 1.8 | 17.1 ± 0.4 | Silt loam | 1.8 ± 0.3 | | 7.7 + 0.0 | | |

[a] Repacked soil layer.

[b] Measured with the pipette-sedimentation method, with fractions: Sand (0.05 mm – 2 mm); Silt (0.002 mm – 0.05 mm) and Clay (< 0.002 mm).

[c] Organic Carbon (OC); Measured by a FORMACS™ HT-i TOC/TN analyser (Skalar , The Netherlands).

[d] Measured with a PDZ Europa ANCA-GSL elemental analyser interfaced with a Sercon 20-22 IRMS with SysCon electronics (SerCon, UK) and EA-IRMS EA IsoLink interfaced through a ConFloIV to a delta Q (Thermo Scientific, Germany). All $\delta^{13}$C values are $^{13}$C/$^{12}$C ratios expressed relative to the international VPDB (Vienna Pee Dee Belemnite) standard.

[e] pH in 1:5 soil:water (volume fraction) suspensions, measured using a glass electrode.

[f] Bulk Density (BD).



## 3 Results

### 3.1 Moisture transport as a function of GWT treatment

#### 3.1.1 Volumetric water content ($\theta_V$) along soil profiles of undisturbed columns

Overall, there was a gradual decrease in $\theta_V$ with increasing height above the GWT for all GWT and soil texture treatments (Fig. 2). The applied water appeared to primarily affect $\theta_V$ in the upper 10 cm and not in deeper soil layers.

In the loamy sand columns, the average moisture level at –85 cm and –60 cm depth was significantly lower (P = 0.016 and P = 0.007, respectively) for GWT –165 cm than GWT –115 cm (Table 2). At –30 cm, only a less pronounced difference (P = 0.067) was observed, while at –10 cm, GWT treatment did not impact $\theta_V$ (P = 0.294). The $\theta_V$ at an equivalent height of about

80 cm above both GWTs was lower in case of the –115 cm GWT treatment at –30 cm compared to at –85 cm for the –165 cm GWT treatment, which was probably the result of evaporative losses. Conversely, for the GWT –165 cm treatment, $\theta_V$ was comparable at –30 cm and –10 cm, and this might indicate limited impact of evaporative losses on topsoil moisture.

In the sandy loam columns, $\theta_V$ generally decreased with increasing height above the GWT, aside from lower $\theta_V$ at 55 cm above the –115 cm GWT (0.237 m3 m$^{-3}$) compared to that at 80 cm above the –165 cm GWT (0.282 m$^3$ m$^{-3}$). This inconsistency

might be explained by the elevated bulk density  at –60 cm (1.75 ± 0.02 g cm$^{-3}$) compared to that at –85 cm (1.65 ± 0.02 g cm$^{-3}$) (Table 1). The $\theta_V$ differed between both GWT treatments at –85 cm (P = 0.012), –60 cm (P = 0.028) and at –10 cm (P = 0.026) but surprisingly not at –30 cm (P = 0.31) (Table 2).

In the silt loam columns, $\theta_V$ was comparable (about 0.310 m$^3$ m$^{-3}$) at depths below 30 cm from the surface (Table 2). The surprisingly lower $\theta_V$ at –120 cm (0.270 m$^3$ m$^{-3}$) for the –165 cm GWT treatment might again be explained by the relatively

higher BD (1.67 ± 0.01 g cm$^{-3}$) at that depth (Table 1). Between the GWT treatments, there were only marginally significant differences in $\theta_V$ at –85 cm (P = 0.062) and –60 cm (P = 0.092), but not at –30 cm (P = 0.160). At –10 cm, $\theta_V$ was lower at GWT –165 cm than at GWT –115 cm (P = 0.028).

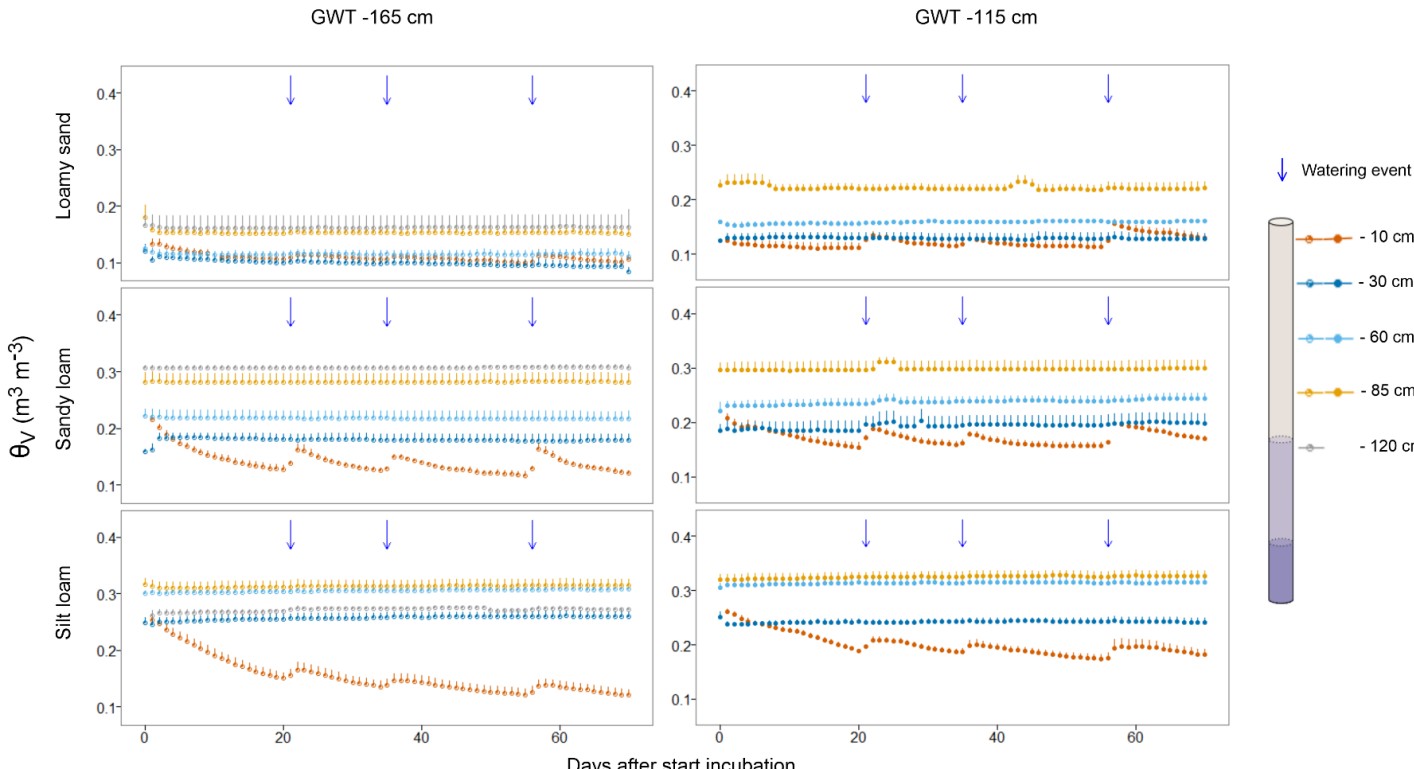


**Figure 2: Evolution of soil moisture (θv) depth profiles (n = 4) over time for two GWT (groundwater table) treatments (depths at –165 and –115 cm) and three textures (loamy sand, sandy loam and silt loam). Note that the GWT treatment –115 cm does not have a sensor installed at –120 cm depth (i.e. below the established GWT).**



**Table 2: Estimated marginal means of moisture contents (θᵥ) measured at different depths as a function of GWT treatment (depths at −165 cm and −115 cm).**

| | Sensor depth (cm) | Relative position above the GWT (cm) | | $\theta_v$ (m³ m⁻³) | |
|---|---|---|---|---|---|
| | | **GWT** | **GWT** | **GWT** | **GWT** |
| | | **−165 cm** | **−115 cm** | **−165 cm** | **−115 cm** |
| Loamy sand | −10 | 155 | 105 | 0.109 | 0.123 |
| | −30 | 135 | 85 | 0.099 | 0.129 · |
| | −60 | 105 | 55 | 0.115 | 0.159 * |
| | −85 | 80 | 30 | 0.153 | 0.222 * |
| | −120 | 45 | / | 0.162 | / |
| Sandy loam | −10 | 155 | 105 | 0.140 | 0.172 * |
| | −30 | 135 | 85 | 0.179 | 0.193 |
| | −60 | 105 | 55 | 0.217 | 0.237 * |
| | −85 | 80 | 30 | 0.282 | 0.298 * |
| | −120 | 45 | / | 0.307 | / |
| Silt loam | −10 | 155 | 105 | 0.153 | 0.200 * |
| | −30 | 135 | 85 | 0.257 | 0.242 |
| | −60 | 105 | 55 | 0.305 | 0.313 · |
| | −85 | 80 | 30 | 0.313 | 0.325 · |
| | −120 | 45 | / | 0.271 | / |

Symbols "·" and "*" indicate that moisture was significantly higher ($P < 0.1$ and $P < 0.05$, respectively) for GWT treatment −115 cm when compared to the −165 cm treatment. Note that GWT treatment −115 cm did not have a sensor installed at −120 cm depth (i.e. under the established GWT).



### 3.1.2 Hydraulic head differences

Across all three textures and both GWT treatments, there was a negative hydraulic head difference (ΔH) between –10 cm and –30 cm during the first few days of the incubation. This indicates gravitational water transport due to wetter repacked topsoil layers at the start of the experiment towards the drier undisturbed soil layers. In the loamy sand columns, the ΔH between –10 cm and –30 cm remained mostly negative in case of the –165 cm GWT treatment during the rest of the experiment (Fig. 3). In contrast, between –60 cm and –30 cm, ΔH ranged from +50 to +198 cm WH, indicating upward water movement. For the

shallower –115 cm GWT, water transport between –30 cm and –10 cm was alternately downward and upward depending on the watering applications (viz. ΔH ranging from –80 to +30 cm WH), followed by slight positive and quite constant ΔH (+50 cm WH) between –60 cm and –30 cm. For both GWT treatments, ΔH between –85 cm and –60 cm remained slightly positive throughout the experiment period. Between –120 cm and –85 cm we found an unforeseen negative ΔH of ~ –95 cm WH.

In the sandy loam columns, between –30 cm and –10 cm ΔH was positive and fluctuated in response to the water applications

for both GWT treatments. In the –165 cm GWT treatment, higher ΔH maxima were observed as a result of drier topsoil compared to the –115 cm GWT treatment. Both GWT treatments exhibited fairly consistent positive ΔH between –60 cm and –30 cm, as well as between –85 cm and –60 cm, with average ΔH values of ~ +37 cm WH and ~ +20 cm WH (GWT –165 cm), and ~ +32 cm WH and ~ +29 cm WH (GWT –115 cm), respectively. Surprisingly, like with the loamy sand columns, we measured a negative ΔH of about –95 cm WH between –120 cm and –85 cm, for the GWT –165 cm treatment, i.e. close to the

GWT.

In the silt loam columns, ΔH in the topsoil increased up to +10800 cm WH in case of the –165 cm GWT treatment as a result of topsoil drying out compared to underlying soil. In case of the shallower GWT of –115 cm, the ΔH between –10 cm and –30 cm alternated around 0 cm WH, with temporary negative values (i.e. downward moisture transport) directly after watering events, then followed by an increase in ΔH to positive values after several days (i.e. upward moisture transport). For GWT –

165 cm positive ΔH values were found in all subsoil layers, which tended to decrease over time from +830 to +450 cm WH, +245 to +100 cm WH, +120 to +30 cm WH and +110 to +90 cm WH between respective depths of –60 cm to –30 cm, –85 cm to –60 cm and –120 cm to –85 cm. For the –115 cm GWT, upward (but rather constant) moisture transport existed as well, with average ΔH values of +1200 cm WH and +135 cm WH, for –60 cm to –30 cm and –85 cm to –60 cm, respectively.



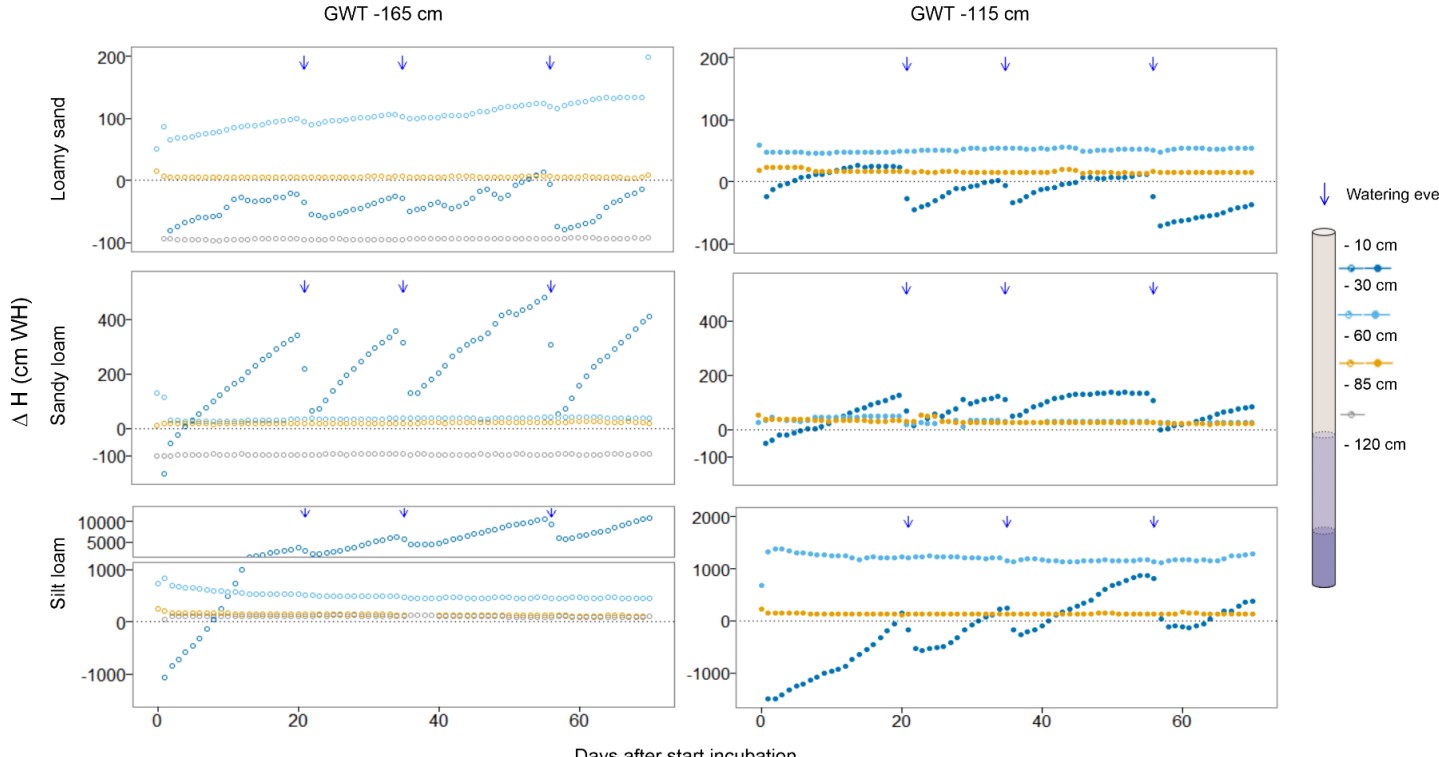

**Figure 3: Difference in hydraulic head (ΔH) over time for both GWT treatments (–165 cm and –115 cm) measured in loamy sand, sandy loam and silt loam soil columns. A positive (negative) ΔH indicates a hydraulic head difference enabling upward (downward) moisture transport. Note that GWT treatment –115 cm did not have a sensor installed at –120 cm depth and therefore no ΔH with –85 cm could be shown.**



### 3.2 Mineralization of added ryegrass

### 3.2.1 Mineralization rates and moisture in topsoil

During the initial five days of the incubation ryegrass mineralization rates were highest with mean maxima of 1643, 1053 and 1133 µg $C_{ryegrass}$ kg$^{-1}$ soil h$^{-1}$ for loamy sand, sandy loam and silt loam columns, respectively. From day seven onwards, the rates decreased gradually over time and after day 30, averaged around 72, 62 and 78 µg $C_{ryegrass}$ kg$^{-1}$ soil h$^{-1}$ for the three respective textures (Fig. 4). There was no significant effect of GWT treatment on the mean $C_{ryegrass}$ mineralization rate across the entire incubation period per texture, although for some individual measurement days, rates did alternate between the GWT treatments. After the watering applications, mineralization rates in the drier, –165 cm GWT treatment, soil seemed to be more sensitive to the moisture input. Significant differences were observed only in comparison to the –115 cm GWT from the second watering application onwards in the loamy sand soil, and after the third application for the sandy loam and silt loam soil.





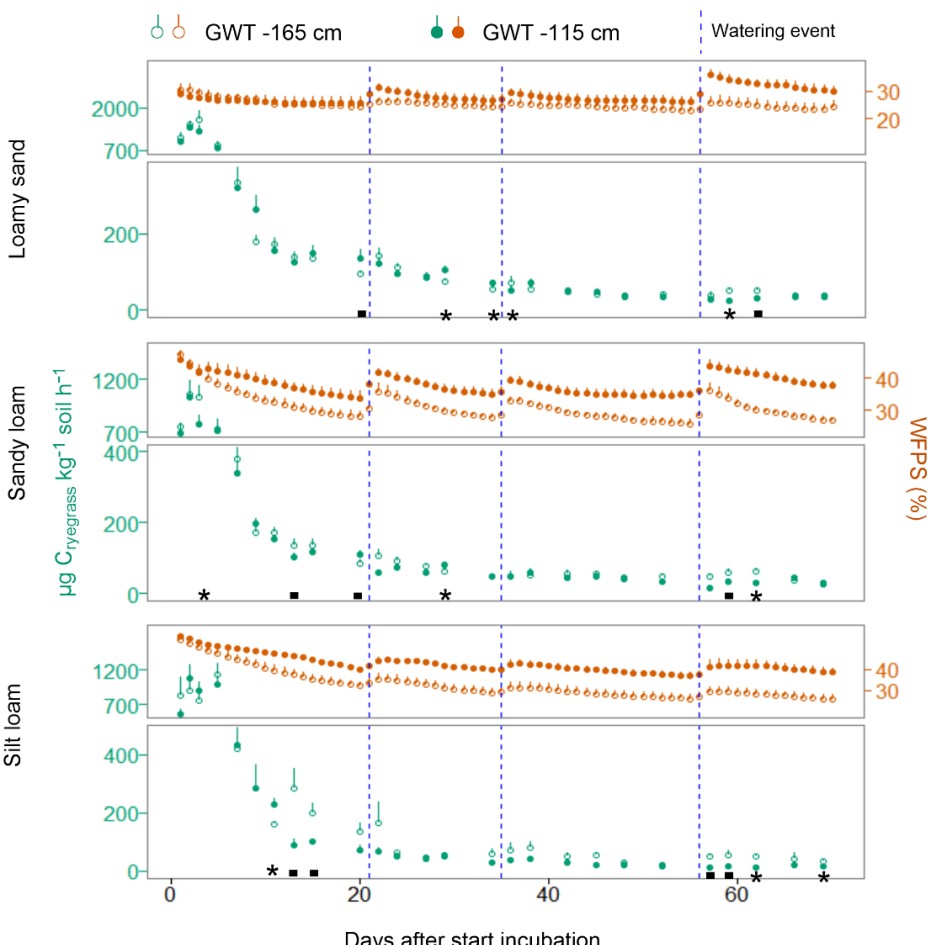

Figure 4: Ryegrass mineralization rates combined with the topsoil moisture content expressed in % of Water-Filled Pore Space (WFPS) for both GWT treatments in the loamy sand, sandy loam and silt loam soil columns (n = 4). Symbols "·" and "*" indicate significant differences for a specific measurement day at $P < 0.1$ and $P < 0.05$, respectively.



### 3.2.2 Cumulative ryegrass mineralization

Overall, 15 – 20 % of the added $C_{ryegrass}$ was mineralized over the course of the 70-days incubation period. Cumulative $C_{ryegrass}$-min did not differ between the three soil textures (19.3 %, 15.7 % and 17.4 % for the loamy sand, sandy loam and silt loam, respectively). In contrast, GWT treatment had a marginal significant effect (P = 0.051) with lower cumulative $C_{ryegrass}$-min in case of the –115 cm GWT (16.2 %) compared to the –165 cm GWT (18.7 %).

The kinetic parallel first-zero-order mineralization model fitted very closely to the cumulative $C_{ryegrass}$-min with NSE-values >
0.98 (Fig. 5). The estimated size of the easily mineralizable pool ($C_f$) was not different between the texture and GWT depth combinations, with average values between 10.3 % and 15.0 % of the initially added amount of $C_{ryegrass}$ (Table 3). The mineralization rate of this fast $C_{ryegrass}$ pool ($k_f$) was not different between the GWT treatments, but it was lower for the silt loam compared to the loamy sand (P = 0.004) and sandy loam (P = 0.049) soils. After the depletion of the fast $C_{ryegrass}$ pool in about 2-3 weeks, the cumulative $C_{ryegrass}$-min further proceeded at a much slower pace following a close to linear course, described by $k_s$. This $k_s$ was significantly lower in case of the –115 cm GWT silt loam soil treatment compared to the other soil texture and GWT depth combinations.



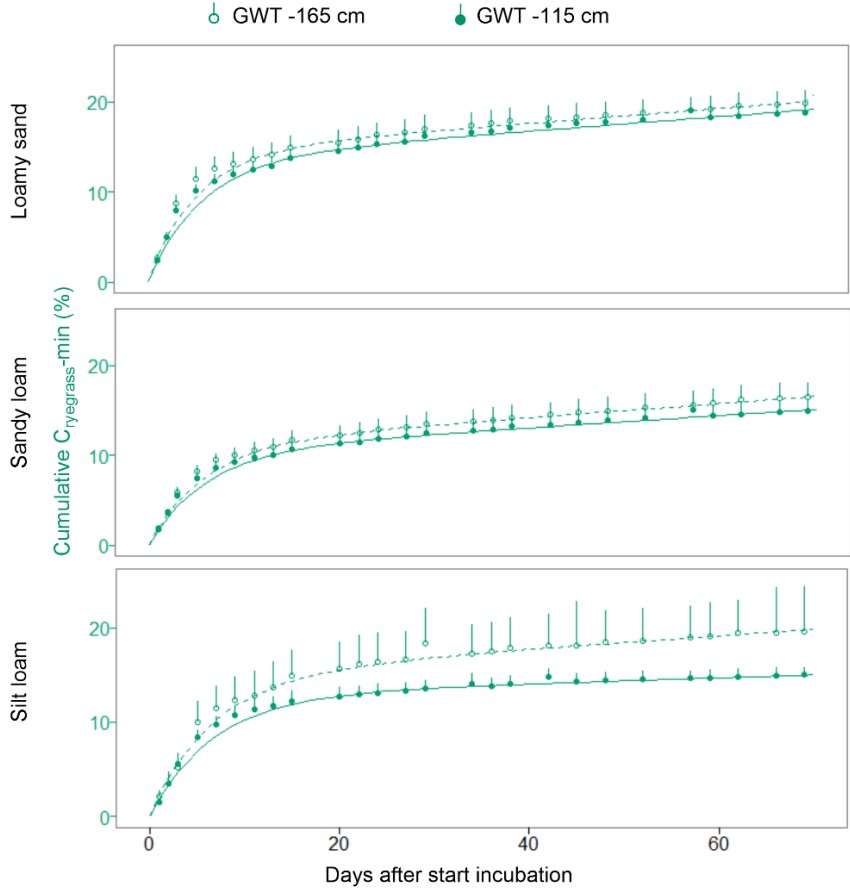

**Figure 5: Cumulative C$_{ryegrass}$-min (%) and fitted parallel first-zero-order kinetic model for both GWT treatments in loamy sand, sandy loam and silt loam soil columns (n = 4).**


**Table 3: Parameters of the parallel first-zero-order kinetic model characterizing the cumulative C$_{ryegrass}$-min for both GWT treatments in loamy sand, sandy loam and silt loam soil columns.**

| | Loamy sand | | Sandy loam | | Silt loam | |
|---|---|---|---|---|---|---|
| | **GWT −165 cm** | **GWT −115 cm** | **GWT −165 cm** | **GWT −115 cm** | **GWT −165 cm** | **GWT −115 cm** |
| C$_f$ (%) | 14.3 ± 1.5 a | 13.5 ± 0.9 a | 11.0 ± 1.0 a | 10.3 ± 1.0 a | 15.0 ± 2.6 a | 12.8 ± 1.0 a |
| k$_f$ (d$^{-1}$) | 0.20 ± 0.00 a | 0.18 ± 0.01 a | 0.18 ± 0.01 a | 0.17 ± 0.00 a | 0.15 ± 0.02 b | 0.15 ± 0.01 b |
| k$_s$ (% d$^{-1}$) | 0.08 ± 0.00 a | 0.08 ± 0.01 a | 0.08 ± 0.01 a | 0.07 ± 0.00 a | 0.07 ± 0.02 a | 0.03 ± 0.01 b |



Letters indicate significant (P < 0.1) differences per parameter between soil texture and GWT depth combinations.



## 4 Discussion

With this experiment, we aimed to infer if and how GWT depth (here either at –165 cm and –115 cm) impacts topsoil moisture
and C mineralization during simulated dry periods that could realistically occur in North-West Europe. Below we discuss
capillary moisture supply as a function of GWT and soil texture combinations (4.1), the impact on OM mineralization (4.2)
and  consequences for modelling of topsoil carbon on the landscape scale (4.3).

### 4.1 To what extent does groundwater table depth affect moisture during simulated drought?

Overall, the established GWT contrast (–165 cm vs. –115 cm) significantly affected topsoil (–10 cm) moisture content in the
sandy loam and silt loam columns, but not in the loamy sand columns (Table 2).

In the loamy sand columns, with a GWT of –165 cm, moisture contents of the shallow layers (–30 cm and –10 cm) were
consistently low (~ 0.1 $m^3/m^3$), and thus situated in the "dry" range of the soil water retention curve (Fig. B1). There was clear
evidence that capillary moisture supply from deeper soil layers towards the topsoil was insignificant for this deepest GWT
treatment. First, although positive hydraulic head differences (ΔH) between –60 cm and –30 cm enabled capillary action, they
displayed an increasing trend throughout the experiment (Fig. 3). Hence, the soil was observed to be drying out at 105 cm and
135 cm above the GWT, indicating clearly that evaporative losses were insufficiently compensated by a capillary water flux.
Second, hydraulic head differences between –30 cm and –10 cm were even negative, excluding upward moisture transport out
of the directly underlying subsoil. In fact, watering events likely caused temporary downward moisture fluxes between these
topsoil layers, as indicated by the fluctuating pattern of ΔH (–30 cm to –10 cm). Considering the moisture retention curve, it
390    emerges that stronger suction forces than the ones recorded here would also not have readily resulted in marked further soil
drying, which explains why moisture at –30 cm remained relatively constant despite temporal changes in ΔH. In sum, the
absence of a significant upward moisture supply to the loamy sand topsoil at deep GWT is most likely directly attributable to
a too limited capillary rise height characteristic of this texture. When the GWT was raised to –115 cm, moisture content at –
30 cm was higher than at the –165 cm GWT, implying that capillary rise markedly impacted soil moisture up to at least 85 cm
above the GWT, less so beyond 105 cm and no more beyond 135 cm.

In the sandy loam and silt loam soil columns, GWT depth did clearly affect the topsoil moisture with higher moisture contents
when the GWT was at –115 cm compared to at –165 cm. Hence, it seems likely that upward moisture supply in the sandy loam
and silt loam columns by capillary rise reached at least up to a height of 135 cm. In case of the silt loam soils, moisture contents
were also consistently high (~ 0.310 $m^3\ m^{-3}$) up to 105 cm above the –165 cm GWT and so the capillary fringe likely extended
beyond 135 cm above the GWT.

Our findings are substantiated by the calculated hydraulic head differences, especially with respect to (dis)continuity of
evaporation from the topsoil layer. For example negative ΔH between –30 cm and –10 cm for the loamy sand during the –165
cm GWT treatment indicated that evaporation ceased, while for the other soil texture and GWT combinations this did not seem
to be the case. However, ΔH was unexpectedly negative between –120 cm and –85 cm for the –165 cm GWT treatment of the





loamy sand and sandy loam columns. These observations imply a downward moisture transport, which seems to be highly unlikely, and we expect that they derive from an accumulation of errors when converting measured $\theta_V$ into H via soil water retention curves, which were obtained as drying curves. Due to hysteresis, such curves can be inaccurate for soil wetting (Hillel, 2003), which was the main expected process at the considered deeper depths near the GWT. A better approach would have been to directly measure hydraulic head with tensiometers, but their installation would likely have strongly disturbed the

soil columns.

In sum, our experiment revealed that upward moisture transport from the GWT to the topsoil occurs when the GWT depth is located within specific depth ranges depending on texture, with a larger impact on moisture supply in case of a shallower (closer to the topsoil) depth. For the loamy sand soils, the depth of the GWT to still affect topsoil via capillary moisture supply

was estimated at 85 cm below the surface. For the sandy loam, topsoil moisture supply occurred for GWT depths up to 135 cm below the soil surface, while for the silt loam soils, it seemed that capillary moisture supply from the GWT also at deeper depths than tested here would still markedly affect the topsoil. Former conducted laboratory experiments observed capillary moisture supply from the GWT up to heights of 117 cm and 149 cm in case of loamy sand soils, and up to 175 cm and 183 cm in sandy loam soils (Malik et al., 1989; Shaw and Smith, 1927). In another study (Lane and Washburn, 1947), upward moisture

transport was seen up to 240 cm for soils with a $D_{10}$ of 0.02 mm, which should resemble our silt loam soil (Mozaffari et al., 2022). These reported results are based on setups with repacked soil columns which do not reflect the soil structure of *in situ* conditions. In contrast, our undisturbed soil columns included small scale heterogeneities, with e.g. macropores, small stones, and cracks as they would occur in the field (Lewis and Sjöstrom, 2010), and this probably explains the somewhat lower capillary rise heights found for loamy sand in our experiment. Our setup was located in a temperature-controlled dark room,

where environmental conditions do not approach the field situation. Ambient outdoor wind, temperature and relative air humidity determine evaporation from the topsoil and thus indirectly co-drive the upward suction force across the soil profile as well (Huo et al., 2020). But overall, we expect that mainly soil texture, structure and GWT depth predominantly dictate the height of capillary rise. Consequently, as we worked with 2 m undisturbed soil columns and realistic GWT depths, we do expect findings of GWT-dependent capillary rise heights and topsoil moisture to be representative for the field situation.

Indeed, when GWT is deeper than the so-called evaporation characteristic length, hydraulic pathways get disconnected and even high evaporative demands will not lead to notable topsoil moisture supply (Balugani et al., 2018; Shokri and Salvucci, 2011). This was also observed in the lighter textured loamy sand soil in our study.

**4.2 Impact of GWT on topsoil OM decomposition**

Soil moisture content ($\theta_V$ or % WFPS) determines soil heterotrophic activity and OM mineralization along a bell-shaped relationship (Manzoni et al., 2012; Moyano et al., 2013; Skopp et al., 1990; Yan et al., 2018). Particularly in the intermediate dry range the response of OM mineralization to soil moisture is strong. Although in the low moisture content range, it is often



more informative to use water potential as units, we chose % WFPS as a directly obtained measure, i.e. without any conversion
through pF curves sensitive to hysteresis. As average topsoil moisture was significantly higher in the –115 cm GWT compared
to the –165 cm GWT treatment for the sandy loam (38 % compared to 31 % WFPS) and silt loam (43 % compared to 33 %
WFPS) soils, we accordingly expected promotion of OM mineralization at shallower GWT for these two soil textures. In
loamy sand, shallower GWT only slightly increased topsoil moisture (28 % vs. 25 % WFPS), and so a minimal effect on OM
decomposition may be expected. Surprisingly however, the GWT treatment induced moisture differences did not hold the
expected effect on $C_{ryegrass}$ mineralization rates (Fig. 4). Moreover, cumulative 70-days $C_{ryegrass}$-min proved lower for the
shallower –115 cm GWT treatment (Fig. 5). However, these observations need to be interpreted with care as GWT treatment
induced topsoil moisture differences only occurred after day 8, while mineralization rates were an order of magnitude greater
during the first five days than later on. Notwithstanding this, about half to two thirds of the cumulative $C_{ryegrass}$-min occurred
early in the first weeks of the experiment. This unwanted asynchrony makes that the effect of GWT treatment on $C_{ryegrass}$-min
should not be evaluated based on the 70-days cumulative $C_{ryegrass}$-min, but rather only after several weeks with GWT treatment
imposed topsoil moisture differences effective by then. From the fitted kinetic model it emerges that the easily mineralizable
$C_{ryegrass}$ pool had already been mineralized around day 14 in case of the loamy sand and sandy loam soil and day 20 for the silt
loam soil. Thereafter, cumulative $C_{ryegrass}$-min followed a relatively constant course, i.e. it was determined by the mineralization
of a more stable $C_{ryegrass}$ pool and the artefact in our experiment makes that evaluation of GWT on topsoil $C_{ryegrass}$ mineralization
should further solely be based on effects on mineralization rate of the more stable $C_{ryegrass}$ pool ($k_s$). We accordingly expected
lower $k_s$ estimates with deeper GWT. However, there was no effect of GWT on $k_s$ for the loamy sand and sandy loam soils,
while for the silt loam soils, it was even lower for the –115 cm than for the –165 cm GWT treatment (0.03 vs. 0.07 % d$^{-1}$,
respectively). These results thus lead to refute the hypothesis that a higher GWT-induced topsoil moisture availability would
promote topsoil C mineralization. To interpret these seemingly illogical effects it needs to be borne in mind that the relation
between soil moisture and C mineralization is complex when moisture fluctuates as in our experiment, compared to when it
remains constant as is typically the case in experiments to infer the bell-shaped $\theta_v$ – C-min relationship. Notably it is well
known that rewetting of dry soil triggers a pulse of microbial activity, resulting in a $CO_2$ flush, and this phenomenon is referred
to as the "Birch effect" (Barnard et al., 2020; Birch, 1958). Former studies found that these respiration pulses enlarge with
bigger change in soil moisture state upon rewetting, but also with increased drier pre-wetting soil conditions (Lado-Monserrat
et al., 2014; Manzoni et al., 2020; Unger et al., 2010; Fischer, 2009; Harrison-Kirk et al., 2013). For example Harrison-Kirk
et al. (2013) observed an increased Birch effect in silt loam soils when pre-wetting moisture state was reduced from 33 % to
22 % WFPS. Indeed particularly shortly after water applications later in our experiment, the rates tended to deviate between
both GWT treatments, with a stronger temporary stimulation in the drier pre-wetting, –165 cm GWT (24 %, 27 % and 29 %
WFPS) topsoil, when compared to the wetter, shallower, –115 cm GWT treatment (26 %, 35 %, 39 % WFPS for respectively
the loamy sand, sandy loam and silt loam soils) (Fig. 4). Accordingly, we postulate that the drier condition of topsoil with
deeper GWT amplified the Birch effect, i.e. rewetting C mineralization pulses caused by the watering events. In summary, the
anticipated enhancing effect of the GWT-induced moisture increase under shallower GWT depth through capillary action on



C mineralization, as represented by $k_s$, was presumably counteracted (in the case of the loamy sand and sandy loam soils) and even overruled (in the case of the silt loam soil) by the stronger Birch effect on drier soils, i.e. soils which were less affected by capillary moisture supply. In other words, the reduced mineralization rate found for silt loam under the shallower GWT ($k_s$)

was the indirect effect of a larger capillary moisture supply compared to the other texture and GWT combinations. As dry-wet cycles are expected to intensify with climate change in Europe, i.e. longer periods of drought followed by intense rainfall events (CEU. JRC., 2022), such GWT-induced control on moisture variations and consequences for the C budget must be carefully considered.

**4.3 Consequences for modelling moisture and SOC balances on the larger spatial landscape scale**

Based on our findings, it emerges that variation in GWT at relatively shallow depths as seen in a large part of our study area (North-West Europe) will contribute to spatial variation in topsoil moisture during periods of limited rainfall. Along, Meles et al. (2020) adapted the often used Topography Wetness Index by inversely weighing it with the GWT depth, resulting into a more accurate index for low-slope landscapes, such as our study region. Ukkola et al. (2016) further reported that there is a

systematic tendency among numerous land surface models to overestimate the consequences of drought. They attributed this to the assumption of a free-draining soil boundary in these models, i.e. no account is taken of a GWT. According to our results, the simplified hydrological modules with free-drainage applied in many soil C models, e.g. DAYCENT (Schimel et al., 2001), BIOME-BCG (Thornton, P.E and Law, B.E, 2010), CERES (Gabrielle et al., 1995), CANDY (Franko et al., 1995), would be less accurate for simulation of topsoil moisture during periods with limited rainfall, as these models do not incorporate capillary

rise in simulating recharge and presuppose that water draining from the soil profile is lost. Just a limited number of biogeochemical models do include bidirectional water flow by defining or even calculating a dynamic GWT, e.g. LandscapeDNDC (Haas et al., 2013; Liebermann et al., 2018) and DAISY (Abrahamsen and Hansen, 2000). However, the question remains regarding the accuracy of such models in simulating upward moisture supply and the relationship between moisture and C mineralization, particularly under dry conditions. With respect to the latter question, Moyano et al. (2013)

concluded that the predictive capacity of current models is still questionable and that these models should incorporate physically-based transport mechanisms for solutes, coupled with a more detailed portrayal of biological reactions to alterations in soil moisture. In order to reproduce respiration patterns caused by phenomena like the Birch effect, Evans et al. (2016) accordingly argued that models should include physicochemical mechanisms linking water content to microbial growth and to diffusion.

Although in our 70-days experiment two GWTs did not suggest a severe impact of GWT on C mineralization, we should not generally conclude that capillary moisture supply is an irrelevant process to be taken along in soil C models. When meteorological droughts extend over longer periods than the simulated period here, the combination of limited moisture supply and increased evaporation might not only lead to soil moisture deficits but it should also diminish groundwater recharge over the longer term (Brauns et al., 2020). This process could then lead to a further reduction of topsoil moisture content, creating



a positive feedback system. As a result, in turn GWT could then deepen further than usually anticipated during summer, which was in fact also eminently observed over the past years in e.g. Flanders, Belgium (VMM et al., 2022). Especially for croplands in North-West Europe with a GWT close to the evaporation characteristic length, a tipping point in their moisture balance may be reached under future expected prolonged periods of drought in spring or summer. We conclude that soil models that are able to simulate upward moisture supply through capillary action and phenomena such as the Birch effect are better positioned

to anticipate soil C trends.

## 5 Conclusion

Variation in GWT depth, typically to occur in North-West European arable land, was found to significantly impact the soil moisture profile of lighter textured soils during periods with limited rainfall. The expected texture dependency of reach of soil moistening by capillary rise could here be quantified to 85 cm above the GWT for loamy sand soils and minimally to 135 cm

for sandy loam and silt loam soils. For situations where the GWT is within these ranges our findings should motivate to include bidirectional water flow, i.e. drainage and capillary rise, in soil models. Nevertheless, contrary to our hypothesis, a rise of GWT did not enhance decomposition of the added substrate (ryegrass). Moreover, it appeared that the magnitude of the Birch effect, i.e. C pulses after rewetting, was inversely affected by the extent of capillary moisture supply during the period of drought. The overall net-effect of the GWT depth on C mineralization is therefore a trade-off and illustrates the complexity of

soil moisture controls on soil biological processes with fluctuating moisture regime. During prolonged periodic droughts, expected to become more frequent under future climate, correct simulation of the mostly neglected capillary rise moisture supply may become imperative for reliable simulation of C cycling in agricultural land in North-West Europe.





**Appendix A**

**Table A1: Comparison of volumetric moisture content measured with sensors at –10 cm and at the start and the end of the incubation of each groundwater treatment ("validation"). Sensor values indicated with "\*" are subjected to a correction via linear regression (see Fig. A2).**

| Location cropland (Texture) | Column replicate | Point in time | $\theta_V$ (m³ m⁻³) GWT –165 cm | | GWT –115 cm | |
|---|---|---|---|---|---|---|
| | | | sensor | validation | sensor | validation |
| **Kruisem** (Loamy sand) | 1 | Start | 0.155 | 0.150 | 0.132 | 0.150 |
| | | End | 0.092 | 0.110 | 0.123 | 0.164 |
| | 2 | Start | 0.139 | 0.150 | 0.122 | 0.150 |
| | | End | 0.125 | 0.097 | 0.126 | 0.155 |
| | 3 | Start | 0.122 | 0.150 | 0.142 | 0.150 |
| | | End | 0.086 | 0.093 | 0.156 | 0.148 |
| | 4 | Start | 0.114 | 0.150 | 0.103 | 0.150 |
| | | End | 0.099 | 0.104 | 0.115 | 0.182 |
| **Bottelare** (Sandy loam) | 1 | Start | 0.205 | 0.218 | 0.106* | 0.218 |
| | | End | 0.122 | 0.137 | 0.092* | 0.156 |
| | 2 | Start | 0.223 | 0.218 | 0.212 | 0.218 |
| | | End | 0.132 | 0.124 | 0.158 | 0.176 |
| | 3 | Start | 0.205 | 0.218 | 0.152* | 0.218 |
| | | End | 0.109 | 0.118 | 0.110* | 0.168 |
| | 4 | Start | 0.225 | 0.218 | 0.145* | 0.218 |
| | | End | 0.121 | 0.117 | 0.103* | 0.162 |
| **Oosterzele** (Silt loam) | 1 | Start | 0.250 | 0.280 | 0.247 | 0.280 |
| | | End | 0.103 | 0.128 | 0.211 | 0.242 |
| | 2 | Start | 0.278 | 0.280 | 0.259 | 0.280 |
| | | End | 0.148 | 0.145 | 0.170 | 0.204 |
| | 3 | Start | 0.241 | 0.280 | 0.266 | 0.280 |
| | | End | 0.125 | 0.146 | 0.176 | 0.170 |
| | 4 | Start | 0.245 | 0.280 | 0.272 | 0.280 |
| | | End | 0.109 | 0.130 | 0.171 | 0.171 |



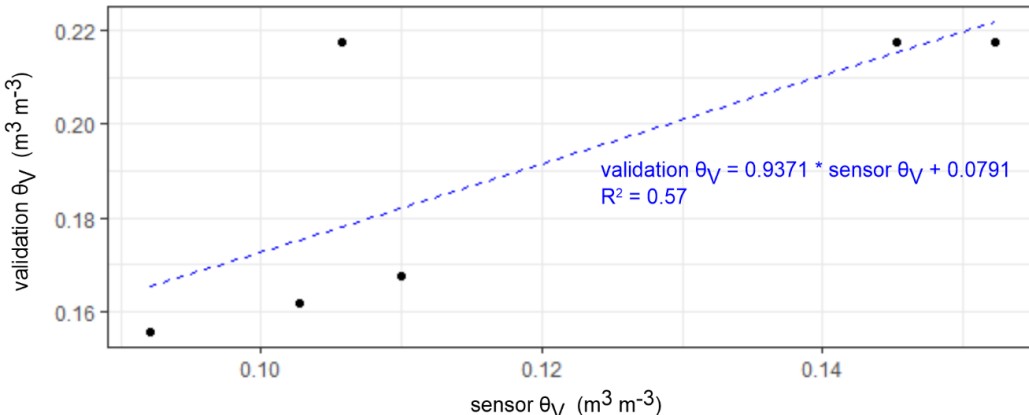


**Figure A2: Sensors installed at a depth of –10 cm in sandy loam columns 1, 3 and 4 during GWT treatment –115 cm were subjected to a correction via linear regression.**



**Appendix B**


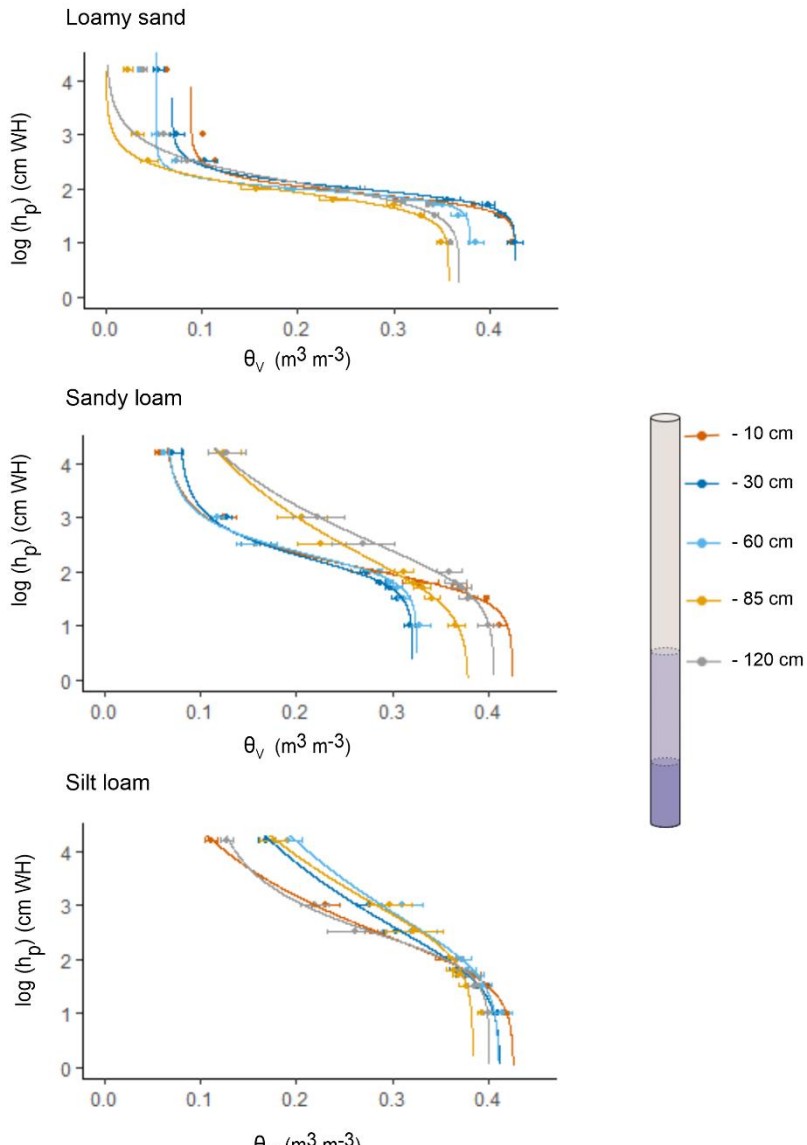

**Figure B1: Soil water retention curves for different depths in our soil columns (n=4; except at −10 cm, n = 2).**



**Data / Code availability**

Raw data and R scripts will be made available upon request.

**Author contribution**

Astrid Françoys: Conceptualization, Methodology, Formal analysis, Visualization, Writing – original draft preparation. Orly Mendoza: Visualization, Writing – review & editing. Junwei Hu: Formal analysis, Writing – review & editing. Pascal Boeckx: Funding acquisition, Writing – review & editing. Wim Cornelis: Funding acquisition, Writing – review & editing. Stefaan De

Neve: Writing – review & editing. Steven Sleutel: Conceptualization, Methodology, Funding acquisition, Supervision, Writing – review & editing.

**Competing interests**

Some authors are members of the editorial board of journal SOIL.

**Acknowledgements**

This study was funded by Flanders Research Foundation (FWO: G066020N). We would like to thank Stijn Willen, Kris Françoys and Viktor Françoys for helping with the technical side of the laboratory setup. Further, we would like to acknowledge the support we got from Patric Buggenhout, Wim De Smet and Hugo Hofman for allowing us to take the soil column samples in their fields.




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
