# Peer review of "The effect of groundwater depth on topsoil organic matter mineralization during a simulated dry summer in North-West Europe"

_EGUsphere, 2024_

## Author Comment (AC2)

*27 replace 'can be' with 'is'*

→ OK

*37 remove 'content'*

→ OK

*50 replace 'moisture' with 'water'*

→ OK

*66 replace 'moisture' with 'water'*

→ OK

*114 with respect to the difficult-to-interpret results of this study, I wonder whether the lab room temperature may have had an effect. As stated, the two GWT tables were imposed for two consecutive time periods. The C turnover is highly sensitive to temperature. Let's assume a more or less standard Q10 of 1.8 for an Arrhenius-type equation. That means for a 10 degrees increase in soil temperature you would have 1.8 times the amount of C turnover. The lab temperature has a standard deviation of plusminus 0.5 degrees. For a 1 degree increase you end up with 18 % more respiration. Now, if the average temperature in the lab for the two consecutive incubation periods differs...*

→ The room where the experiment took place was held under a ventilation system, and the system did not change between the GWT treatments. Therefore, the average room temperature between both treatments was 20.8°C. However, there was indeed some fluctuation in temperature over time during both treatments, hence the standard deviation of 0.5°C. We believe that this variation in room temperature had a smaller impact on C mineralization than the moisture regime itself. Although there was no difference in mean air temperature between the both incubation batches, we do agree that such fluctuation could have impacted our results. Therefore, we particularly agree to be more cautious with the explanatory statements regarding the Birch effect, as they are based on relatively small increments in ryegrass-C mineralization. We will phrase these statements more conditionally and carefully in our discussion and conclusion.

*144 replace 'moisture' with 'water'*

→ OK

*145 replace 'moisture' with 'water'. Well, there is a little issue with terminology: there is either 'water content' or 'moisture'. Please correct for the entire manuscript...*

→ OK

*207 well, the more correct term would probably be 'potential' instead of 'height'*

→The matric head is the height of a water column that corresponds to the matric potential, typically expressed in units of cm water height (cm WH). Thus, 'matric potential' is a general term describing the energy state of water in the soil, with 'cm water height' as its unit of measurement. Therefore, we consider our text correct and leave it as it is.

*209 I suggest to replace 'moisture transport' with 'water flux'*

*209 and 210 replace 'transport' with flux*

→ OK, we adapted the text the following way:

209: The latter were used to calculate hydraulic head differences (ΔH) between two adjacent sensor positions above the GWT, with hydraulic head the sum of matric head and gravitational head. They were used as an indicator for the water flux direction: positive ΔH values indicate a net upward (capillary) water flux, while negative ΔH values signify an overall downward water flux.

*243 I would prefer to replace 'moisture transport' with 'variably-saturated water flux'*

→ We doubt if that would be a correct phrasing, as 'variably-saturated water flux' does not appear to accurately describe the water state and flux in the unsaturated soil (above the groundwater table). However, we propose to adjust the original text:

3.1 Moisture transport as a function of GWT treatment

Into:

3.1 Soil moisture dynamics in response to GWT treatment

*245 that first sentence can be deleted*

→ We think this is an important first impression describing the results. Therefore we would like to keep it. It also includes some information regarding the infiltration depth of the wetting events (which is questioned in further remarks).

*246 Are measured (saturated) hydraulic conductivities available? In combination with the measured soil water retention functions this would allow to estimate the infiltration depth, or*

*even a soil water content profile over depth. To get an idea of how much the upper 20 cm with the labeled C were actually affected by the irrigation for the different soils...*

→ From the measured volumetric water contents (Fig. 2) we could in fact already clearly distinguish that wetting events lifted soil moisture at −10 cm depth, but not at −30 cm depth. So the infiltration depth must be located somewhere between both depths for all three textures. No hydraulic conductivity measurements were taken in these repacked topsoils, so we cannot further define the specific depth of infiltration. However, our main aim was not to assess the impact of these wetting events, but rather to compare the effect of the GWT on topsoil moisture within distinct soil textures, whilst also mimicking true field conditions, i.e. with minimal rainfall representing a dry summer. As the wetting events were identical between both GWT treatments, this approach seems valid to us.

*Figure 2 The y-Axis of graphs should basically always start from zero.*

→ We agree and have adjusted the Y-axis of Figure 2 to start from zero.

*295 A difference in heads is usually referred to as a 'hydraulic gradient'.*

→ We do think there is a difference between a hydraulic head gradient and the hydraulic head difference. The latter is obtained by subtraction of the two hydraulic heads at the two depths and is put in the units cm WH. The gradient, on the other hand, is obtained as the quotient between a hydraulic head difference and the distances between the measuring points, and has units $cm\ cm^{-1}$ or alike. In addition, a 'gradient' would rather suggest a monotonous change in hydraulic head between at least two depths, but it is quite unlikely that we should encounter such trend in our heterogenous soil profiles. Therefore, we opted to just more directly refer to 'hydraulic head differences between adjacent sensors'.

*315 well, now you end up with two factors affecting (or not affecting) the water content in the upper 20 cm with labelled C. The ground water table AND the irrigation. The irrigation will affect the respiration of the different soils differently. This might be difficult to interpret in the end.*

→ We did not compare ryegrass-C mineralization between the different textures. Instead, we focused on the effect of GWT-induced moisture differences within identical soil textures. Additionally, statements about water fluxes were texture-specific, as irrigation might have had different effects on the moisture balance of different textured topsoils. Obviously, the wetting events influenced the topsoil moisture, but we deem them essential to mimic field conditions, where going 70 days without rainfall would be highly unlikely. The small amount of added water mimicking rainfall was based on a realistic scenario, representing a dry summer in the study area, and this regime was kept identical across different textured soils. In fact this scheme also mirrors the real field situation as the collected

soils are in fact often at just short distances of each other in Flanders. This approach then allowed us to assess the net effect of GWT depth.

*Fig. 3 Maybe a log10 scale applied to the y-axis helps this figure*

→ Applying a log scale to negative values is not possible.

*Fig. 4 I am quite sure a log scale applied to the y-axis benefits this figure*

→ We tried a log-scale Y-axis. However, then the positive standard errors of the ryegrass mineralization rates for sandy loam soil resulted in both negative and positive log values due to the original values ranging from <1 to >1. This looks rather confusing, as the displayed errors would represent only the positive standard errors. Therefore, we decided to keep the current scale for the axes.

*379/380 delete or move to results*

→ OK, deleted

*387 I assume '-30' and '-10 cm' are the coordinates in the column (i.e. the depth) and not the difference in pressure head?!*

→ Indeed. If we mention pressure heads, the unit is mentioned as 'cm WH' instead of 'cm' solely. To make that more clear in this specific case, we would now add the word 'depth':

386: Second, hydraulic head differences between –30 cm and –10 cm depth were even negative, excluding upward moisture transport out of the directly underlying subsoil.

*406 No, that is not unlikely. This could be a result of macropores.*

→ We recognize that macropores can serve as preferential flow paths, facilitating a downward water flux through the soil pore network. However, in combination with the volumetric soil moisture data, we think it could not have been a dominant avenue for moisture transport. Higher measured water contents for the –115 cm GWT treatment compared to the –165 cm GWT treatment demonstrate that the net transport of moisture must have predominantly been upwards, as there were no differences in water regime and thus potential downward leaching through macropores between both GWT treatments.

*409 well, there are very small lab tensiometers on the market.*

→ In retrospect, installing compact tensiometers would have certainly been worthwhile compared to only using volumetric moisture sensors. However, we particularly opted for TEROS 10 and TEROS12 sensors (from Meter Group) as these consist of fine, sharp rods. This allowed to pierce our soil columns and the covering waterproof foil with minimal disturbance to the surrounding soil, and minimizing moisture and air losses. The type of small tensiometers with a sharp shaft diameter of less than 1 cm with a high measuring accuracy, for example the TEROS31 from Meter Group, was only released in 2021, the same year of the start of this experiment. Additionally, in previous research we experienced that with tensiometers, there is a higher chance of malfunction compared to volumetric moisture sensors as they must always be filled with degassed and demineralized water and no signal is given when they are not filled properly. Installing both sensor types (volumetric moisture sensors and small lab tensiometers) would have been ideal to provide us with the most complete and reliable data, but it would also have resulted in twice the disturbance and more soil volume being occupied by the sensors in the columns.

*421 Did you observe macropores during sampling?*

→ When opening the soil columns after the experiment, we did not visually detect any preferential flow paths in the soil cores. However, since macropores can be as small as 0.08 mm, we cannot guarantee that absolutely none were present.

*435 these 'bell-shaped' curves were usually determined for disturbed soil samples. And in the wet range the decrease of respiration is rather related to an oxygen deficit and is actually not related to water content. For undisturbed samples the moisture sensitivity function of respiration actually looks different.*

→ The top 20 cm of the soil columns was amended with ryegrass and repacked on top of the undisturbed soil columns (180 cm length). Thus, the soil in which we followed the ryegrass C-mineralization was in fact disturbed, representing the ploughed topsoil in the field.

*438 better state that the microbes show a more direct reaction to the water-filled pore-space than to the pressure head*

→ We are not convinced that microbes generally react more directly to WFPS than to matric potential. To our knowledge, it depends on the specific moisture range. In very dry conditions, water potential is often a better predictor of C mineralization because small decreases in water content can translate in significant changes in pF. We would acknowledge this in our text and explain our rationale for still using WFPS as follows:

438: Particularly in the intermediate dry range the response of OM mineralization to volumetric soil moisture is strong, while at lower water content, mostly water potential is a better predictor. Given the occurring moisture range and to avoid conversion of measured volumetric moisture through pF

curves sensitive to hysteresis, we further used % WFPS as a more directly obtained measure of soil moisture.

*448 I suggest to delete 'unwanted asynchrony'*

→ We propose to alter the following original text:

448: This unwanted asynchrony makes that the effect of GWT treatment on $C_{ryegrass}$-min should not be evaluated based on the 70-days cumulative $C_{ryegrass}$-min, but rather only after several weeks with GWT treatment imposed topsoil moisture differences effective by then.

into:

448: This implies that the effect of GWT treatment on $C_{ryegrass}$-min should not be evaluated based on the 70-days cumulative $C_{ryegrass}$-min, but rather only after several weeks, with GWT treatment imposed topsoil moisture differences effective by then.

*453 which 'artefact'?*

→ The artefact refers to the delayed emergence of topsoil moisture differences induced by the GWT treatment, which became apparent only after day 8 of the incubation. In contrast, the peak in mineralization of the easily degradable ryegrass-C pool occurred within the first few days. We can see that this term artefact was rather unclear and therefore we propose to alter the text from:

450: From the fitted kinetic model it emerges that the easily mineralizable $C_{ryegrass}$ pool had already been mineralized around day 14 in case of the loamy sand and sandy loam soil and day 20 for the silt loam soil. Thereafter, cumulative $C_{ryegrass}$-min followed a relatively constant course, i.e. it was determined by the mineralization of a more stable $C_{ryegrass}$ pool and the artefact in our experiment makes that evaluation of GWT on topsoil $C_{ryegrass}$ mineralization should further solely be based on effects on mineralization rate of the more stable $C_{ryegrass}$ pool ($k_s$).

into:

450: From the fitted kinetic model it emerges that the easily mineralizable $C_{ryegrass}$ pool had already been mineralized around day 14 in case of the loamy sand and sandy loam soil and day 20 for the silt loam soil. Thereafter, cumulative $C_{ryegrass}$-min followed a relatively constant course, i.e. it was determined by the mineralization of a more stable $C_{ryegrass}$ pool, and therefore the evaluation of GWT on topsoil $C_{ryegrass}$ mineralization should further solely be based on effects on mineralization rate of the more stable $C_{ryegrass}$ pool ($k_s$).

*457 please replace 'refute' with 'reject'*

→ OK

*458 replace 'it needs .. in mind' with 'against the background'*

→ We propose to alter the text from:

458: To interpret these seemingly illogical effects it needs to be borne in mind that the relation between soil moisture and C mineralization is complex when moisture fluctuates as in our experiment, compared to when it remains constant as is typically the case in experiments to infer the bell-shaped $\theta v$ – C-min relationship.

To:

To interpret these seemingly illogical effects, it is important to consider the complexity of the relationship between soil moisture and C mineralization, especially when moisture fluctuates as it did in our experiment. This is in contrast to typical experiments to infer the bell-shaped $\theta_V$–C-min relationship where moisture is held constant.

*500 but not only the GWTs were modified, simultaneously precipitation was simulated*

→ Yes, but with identical regimes of timed wetting events applied to both GWT treatments, the difference in both experimental batches is still solely the groundwater table depth. We would therefore keep the original phrasing.

*516 now, if Richards equation is used there will always be water flux in all directions. It might be constructive to advocate Richards-based models in contrast to bucket models...*

→ Although both mentioned model types use the Richards equation, free-drainage soil models only simulate capillary moisture transport towards adjacent unsaturated soil layers. They do not include upward moisture transport from the groundwater into the unsaturated zone. Therefore, we think this contrast in both approaches is of particular relevance for this paper as we observed variable influence of the groundwater on the soil water profile for the investigated texture and GWT depth combinations. As a compromise, we propose to now explicitly mention tipping-bucket models shortly in the discussion:

484: Ukkola et al. (2016) further reported that there is a systematic tendency among numerous land surface models to overestimate the consequences of drought. They attributed this to the assumption of a free-draining soil boundary in these models, i.e. no account is taken of a GWT. According to our results, simplified hydrological modules using the Richards equation but with free-draining lower boundaries applied in many soil C models, e.g. DAYCENT (Schimel et al., 2001), BIOME-BCG (Thornton, P.E and Law, B.E, 2010), CERES (Gabrielle et al., 1995), CANDY (Franko et al., 1995), would be less

accurate for simulation of topsoil moisture during periods with limited rainfall, as these models do not incorporate capillary rise in simulating recharge and presuppose that water draining from the soil profile is lost. The discrepancy with real in-soil occurring physical processes becomes even larger for models that not use the Richards equation but instead employ a simplistic cascade bucket approach, such as DSSAT (Jones et al. 2003). Only a limited number of biogeochemical models based on the Richards equation also include bidirectional water flow between the saturated and unsaturated zones by defining or even calculating a dynamic GWT, e.g. LandscapeDNDC (Haas et al., 2013; Liebermann et al., 2018) and DAISY (Abrahamsen and Hansen, 2000). However, the question remains how accurate such models simulate upward moisture supply and relate C mineralization to moisture content, particularly under dry conditions.

516: For situations where the GWT is within these ranges our findings should motivate to include bidirectional water flow between the saturated and unsaturated zone, i.e. drainage and capillary rise from the groundwater, in soil models.

---

## Author Comment (AC3)

*Abstract:*

*L21 this is the first location where wetting events are mentioned. Please describe these events earlier in the abstract (L15)*

→ We actually already mention them in line 17, but there we referred to 'water applications' instead of 'wetting events.' To avoid confusion, we propose to revise our phrasing:

L17: We examined (1) moisture supply by capillary rise along the soil profile and specifically into the top 20 cm soil, and (2) consequently the effect of GWT on decomposition of an added $^{13}$C-enriched substrate (ryegrass) over a period of ten weeks, with limited wetting events representing a dry summer.

*L24 this sentence is quite long and complicated, consider rewording for clarity or breaking up into 2 shorter sentences.*

→ We agree and would adapt the sentence the following way:

L24: We postulate that the Birch effect might have been magnified following the rewetting of drier topsoils under deeper GWT levels. This then resulted in enhanced mineralization compared to conditions where the soil remained consistently wetter with shallower GWT level.

*L30 the last two words in the abstract are the birch effect. For anyone who is not familiar with it this is a very confusing ending to an abstract. I suggest either explaining the birch effect at the top of the abstract or describing the process (i.e. enhanced mineralization after wetting of dry soil) instead of saying birch effect.*

→ The experiment was not set up with the idea to specifically asses the Birch effect, therefore we would not want to start the abstract with a definition of this phenomenon. We do agree that some further clarification is needed before employing the term 'Birch effect' in the abstract.

We propose to alter the current abstract text from:

L24: We postulate that the Birch effect might have been magnified following the rewetting of drier topsoils under deeper GWT levels. This then resulted in enhanced mineralization compared to conditions where the soil remained consistently wetter with shallower GWT level.

To:

L24: We postulate that rewetting might have induced a stronger mineralization response, often referred to as the Birch effect, in drier topsoils compared to conditions where the soil remained consistently wetter with shallower GWT level.

And from:

L28: However, the net effect on topsoil C mineralization is complex and correct simulation of C mineralization may require further integration of specific processes connected to fluctuating soil moisture state, such as the Birch effect.

To:

L28: However, the net effect on topsoil C mineralization is complex. A correct simulation of C mineralization may require further integration of specific processes connected to fluctuating soil moisture state, such as the increased mineralization response after rewetting.

*Intro:*

*In general, I think the phenomenon of capillary rise in drying soils is known and as the authors note has been thoroughly studied in the context of water availability etc. What is new here is the effects of this phenomenon on C mineralization, given the effects of water and changes in water content on it. The intro should therefore, I think, focus more on this part and here's a good place to describe the birch effect and fit it into your hypotheses, and less on what is known and why your method has merits over other methods.*

→ We agree with the suggestion to replace a few sentences about previous research regarding capillary rise in terms of water availability with a hint already towards real fluctuating moisture regimes and the Birch effect. We would, however, rather not include the Birch effect in our hypothesis, as the experiment was not set up with the idea to asses this specifically. At the end of the document (in response to the last comment) a proposal for the adapted sentences of the introduction can be found.

*L34 a word is missing before less. Becomes? Is?*

→ OK, we rephrased it the following way:

L34: When soil desiccates, soil-water potential becomes strongly negative, making eco-physiological conditions for soil micro-organisms less favorable.

*L34 In particular, (add comma)*

→ OK

*Methods:*

*In general, the methods are appropriate for the proposed research questions and are well described. A few comments:*

*L105 a layer of silt clay loam was added between the two 1m segments to ensure connectivity. I understand the reasoning for this given that two cores are artificially places on top of the other, but doesn't this bias the whole capillary rise measurement?*

→ The 1 m segment cores were sampled separately due to the limiting length of the auger (L = 100 cm). The rationale for adding this silt clay loam soil was indeed to ensure a good hydraulic connection between the two segments. In retrospect, it would have been better to sample additional local soil from a depth of approximately 100 cm and use that instead. However, in both GWT treatments, the same setup was used, and observed decreases in $\theta_V$ with increasing height above the GWT were very logical. This suggests that overall any potential artefact effect of e.g. water absorption in the connecting soil layer must have been limited.

*L114-L115 Because two GWT treatments were consecutively applied to each core (all in the same order of GWTs) it means that the second GWT treatment is not independent from the first because it carries over the effects of having been subjected to the first GWT treatment. Can you comment on this. If random cores were given the reverse order of treatments and shown that this does not affect the results that would have been compelling.*

→ We deliberately handled the deepest GWT treatment first. This approach ensured that any potential impact of moisture transport on soil structure was confined to a height above the GWT, which was then exceeded during the subsequent, shallower GWT treatment. Between the two GWT treatments, the barrels with the water representing the GWT were completely emptied, and the soil columns were left to dry out to a condition analogous to the initial state of the first GWT treatment. We will add a sentence to the Materials and Methods section to clarify this approach further. Alternatively, we could have sampled 24 cores and both GWT treatments conducted simultaneously. However, the collection of such long intact soil cores and preparing them in 2 m long setups was labor-intensive. In fact the 4 used cores per texture were already selected out of a larger collected set where the non-used cores displayed damage incurred during sampling and further handling. Hence these practicalities constrained using a fully randomized approach. Nevertheless, by conducting the treatments sequentially, we were able to precisely assess the effects of the two treatments on identical columns (pairwise comparison). This approach eliminated the potential influence of heterogeneity in soil, which could have been an affecting factor on the moisture dynamics. We do want to point out that we only assessed mineralization in the topsoil, which was in fact replaced going from the first deeper GWT batch to the second shallower GWT batch.

*L121 I was disappointed that the authors did not report their results for the mineralization of native SOC and only show the 13CO2 results. This would have given a fuller picture of C dynamics in their systems. Also, because the added 13C litter is likely to 'exist' as particulate organic matter and a large proportion of SOC 'exists' as mineral associated organic matter,*

*quantifying their relative mineralization under different GWT treatments could have greatly improved your hypotheses and results, especially given the lack of difference in 13C mineralization between soils. Since that data surely exists and does not require repeating or doing new experiments, I highly encourage the authors to include this data. The authors ` concerns regarding soil properties effects on SOC mineralization can be partially addressed by normalizing CO2 to C content, and besides the mineralization of the 13C litter is likely also impacted by various properties.*

→ $CO_2$ efflux from native SOC was higher during the first treatment (deeper GWT with lower topsoil $\theta_V$) compared to the second treatment (shallower GWT with higher $\theta_V$) for all textures. This counterintuitive outcome suggests that a substantial portion of the native $CO_2$ efflux resulted from SOC from the undisturbed and non-renewed part below 20 cm. Therefore, if we intended to compare native SOC mineralization, we should have used completely fresh soil columns for both treatments. As mentioned in our response to the above previous referee comment, we prioritized assessing the effect of GWT depth on identical columns to eliminate the potential impact of soil heterogeneity on moisture dynamics. This setup trade-off was initially decided, and is also the very reason why we worked with the $^{13}$C-labeled ryegrass.

*L132 Why did you use VWC sensors which have to be soil-calibrated and then have to be converted to matric potentials using a retention curve instead of using water potential sensors?*

→ We preferred the VWC sensors because of their design. They consist of very fine, sharp rods, which allowed to pierce the intact soil columns and waterproof foil with minimal disturbance to the surrounding soil, while minimizing moisture and air losses. Additionally, in previous research we experienced that with tensiometers, there is a higher chance of malfunction compared to volumetric moisture sensors as they must always be filled with degassed and demineralized water and no signal is given when they are not filled properly. Installing both sensor types (volumetric moisture sensors and laboratory tensiometers) would have been ideal to provide us with the most complete and reliable data, but it would have also resulted in twice the disturbance and more soil volume being occupied by the sensors in the columns.

*L180 delta13C of CO2 undergoes fractionation at different diffusivities (e.g. water contents). https://agupubs.onlinelibrary.wiley.com/doi/full/10.1029/2008JG000766*

*Given that your labeled material wasn't very highly enriched, such differences can have an effect on your calculations. Were the parallel incubations of end members (L179-180) done at the same GWTs as the experimental incubations?*

→ We assessed the isotopic signature of $CO_2$ emitted from native SOC and ryegrass using parallel 20 cm packed soil columns, so not in the 2 m soil columns, as already described in L180-186. Working with soil on top of undisturbed 180 cm soil columns in GWT barrels, would have been complicated as there would also be 'contamination' of $CO_2$ being emitted from the underlying soil, limiting to precisely assess the $\delta^{13}C$ of $CO_2$ derived from ryegrass mineralization. In these ancillary soil core incubations care was taken to as closely as possible simulate the moisture regime as it occurred in the main experiment: viz. every few days the 20 cm soil columns were weighed, and (a small amount of) water was added if their $\theta_V$ became lower than certain thresholds (0.1, 0.12 and 0.15 $m^3$ $m^{-3}$ for respectively loamy sand, sandy loam and silt loam soils), while also the larger wetting events were applied identically as in the 2 m columns. We would not further comment on this in the revised version.

*Results:*

*Consider renaming the treatments to something friendlier on the eye of the readers (e.g. instead of -165 cm and -115 cm GWT, GWT-deep and GWT-shallow)*

→ We always explicitly mention the GWT depths because it is crucial for understanding the extent of the capillary moisture transport. Using terms like "deep" and "shallow" might not prompt readers to look up the specific depths to which we are referring. We also think these are terms rather open to interpretation. In the results and discussion section, we did regularly refer to the -115 cm GWT as the "shallower -115 cm GWT treatment" (L301, L313, L443, L446…). We propose to now also add this "shallower" and "deeper" term explicitly to the figures and tables (but then in addition to -165 cm and -115 cm).

*Discussion:*

*L445-450 I agree that the asynchrony between 13C mineralization and water content differences are difficult to overcome. However, the differences in cumulative mineralization (fig5) and rate (fig4) become significant only as WFPS differences become significant. So I am not convinced that this surprising result is because the Birch effect was a more dominant process than water content. Could it be that litter 13C mineralization was favored at lower WFPS because it was occluded within pores that still retained water at lower WFPS especially in silt loam (e.g., https://doi.org/10.1016/j.soilbio.2022.108777), while other C sources were preferentially mineralized at higher WFPS because they were in larger pores? I again encourage the authors to look at the native C mineralization to provide a clearer picture of C mineralization in your experiments.*

→ Although we did not show the SOC mineralization results, we found higher $CO_2$-SOC for the -165 cm GWT treatment, or when the topsoil was drier for sandy loam and silt loam. This is the opposite of your suggestion for silt loam soils ('preferential mineralization of other C sources than the ryegrass at higher WFPS'). However, as mentioned earlier, we cannot distinguish clearly which part of this $CO_2$-

SOC efflux is coming from the renewed topsoil, and which part is from the undisturbed, non-renewed 180 cm soil column, which could already have been partly depleted for the second GWT treatment. We carefully considered the Birch effect because ryegrass-C mineralization rates were significantly higher mainly after the third wetting event for all three textures. We expect that the occlusion of the added ryegrass was limited within the relatively short incubation period of 70 days, but such could only be confirmed by soil physical fractionation and we would argue that such extra work is beyond the scope of the current study. Alternatively, it could perhaps also be argued that in silt loam soil, in particular under the shallow -115 cm groundwater, soil conditions for ryegrass mineralization already became a bit too wet (although still just around 40 % WFPS), i.e. with lesser availability of $O_2$ limiting mineralization. Even though we had considered such explanations, we did, however, not, bring them up in the discussion as without metrics of aeration, $O_2$ concentrations or redox potential, such an interpretation is speculative. Should the editor and referee see this nevertheless fit we could briefly complement the discussion with such alternative potential interpretations of the found results.

*Regardless, if the Birch Effect turns out to be such an important aspect of the discussion of the C mineralization results, it should be explicitly termed, explained, and integrated in your hypotheses in the introduction.*

→ We did not initially emphasize the Birch effect because we actually did not anticipate it as a potential explanatory factor for our results. To do so posteriorly would not be entirely appropriate. Our primary focus was on the impact of the GWT through capillary moisture transport, which is why we concentrated the introduction on this aspect. However, we agree that we could replace a few sentences (we propose to delete 6 lines) about previous research regarding capillary rise with information about C mineralization and moisture regimes under real, fluctuating conditions, while also hinting at the Birch effect already (adding 6 new lines). Therefore, we propose the following adaptation to the 
[revised manuscript text omitted]

---

## Author Response (AR2)

Dear editor,

We regret that one of the referees was disappointed with our first revision. We kept proposed changes to the text at times limited to maintain the manuscript's clarity. Finding the balance between providing more detail (as often requested by referees) vs. keeping a manuscript text fluid and comprehensible is always somewhat of a challenge. In hindsight, we apparently misjudged this and apologize that reviewer 2 's concerns were insufficiently acknowledged. We think that our elaborate responses to each of the three referees demonstrate that we do take referee advice seriously. We were thus surprised by referee 2's decision to advise 'rejection' of our manuscript and would have hoped to have gone further in dialogue. We are grateful the editor allowed us to do so and now propose to make further revisions to the manuscript and provide extra details on comments raised by reviewer 2 and two by reviewer 3.

Below, we re-list the *major remarks of the referees*, along with updated author responses, as well as explicit the revised sections of the manuscript (in green, with specifically altered parts underlined). Grammar-related remarks or remarks that have already been addressed elaborately during the first revision (including revised text in the manuscript) have been omitted from this list.
* * *
**RC2**:

*114 with respect to the difficult-to-interpret results of this study, I wonder whether the lab room temperature may have had an effect. As stated, the two GWT tables were imposed for two consecutive time periods. The C turnover is highly sensitive to temperature. Let's assume a more or less standard Q10 of 1.8 for an Arrhenius-type equation. That means for a 10 degrees increase in soil temperature you would have 1.8 times the amount of C turnover. The lab temperature has a standard deviation of plusminus 0.5 degrees. For a 1 degree increase you end up with 18 % more respiration. Now, if the average temperature in the lab for the two consecutive incubation periods differs...*

➔ Firstly, a Q10 of 1.8 implies that per 1°C increase in temperature, respiration rises by 8 %, not 18 %. Notwithstanding, we acknowledge that by the batch design of our experiment it is important to understand whether or not differences in temperature could have instead explained part of the observed contrast in respiration between the GWT treatments. Average room temperatures were identical between the consecutive GWT treatments, but we temperature did fluctuate. We propose to now **further specify the average temperature and standard deviation during both GWT treatments separately** in the M&M:

L120-121: The setup was installed in a temperature controlled (20.8 ± 0.7°C during GWT −165 cm and 20.8 ± 1.5°C during GWT −115 cm) dark room.

➔ **We believe** that despite the fluctuations, **the moisture regimes still had a significant impact onto ryegrass mineralization**. Particularly so in the silt loam soils, where the GWT treatment had the greatest effect on topsoil moisture due to capillary rise. Accordingly, the mineralization

rate of the more stable $C_{ryegrass}$ pool ($k_s$) was significantly lower under the −115 cm GWT treatment compared to the −165 cm GWT treatment in these silt loam soils. In contrast, for other soil textures where similar air temperature fluctuations occurred but capillary rise had less influence on topsoil moisture, the $k_s$ values remained consistent across both GWT treatments. Furthermore, after the third wetting event, mineralization rates in silt loam soils differed substantially between the GWT treatments with mineralization rates of 51 vs. 13 µg $C_{ryegrass}$ kg$^{-1}$ soil h$^{-1}$ (on day 57), or a 292.31 % increase in respiration. This is a lot higher than the suggested increase the T fluctuations would cause. In the **results** section, we propose to **include data of the specific mineralization rate values** to provide readers with a **clearer understanding of the magnitude** of differences between the GWT treatments, which may not have been fully apparent in the original figure due to its broad scale:

L352-356: After the watering applications, mineralization rates in the drier, −165 cm GWT treatment, soil seemed to be more sensitive to the moisture input. Significant differences were observed only in comparison  to the −115 cm GWT from the second watering application onwards in the loamy sand soil (72 vs 51 µg $C_{ryegrass}$ kg$^{-1}$ soil h$^{-1}$ at day 36), and after the third application for the sandy loam (59 vs 32 µg $C_{ryegrass}$ kg$^{-1}$ soil h$^{-1}$ at day 59) and silt loam (51 vs 13 µg $C_{ryegrass}$ kg$^{-1}$ soil h$^{-1}$  at day 57) soil.

➔ However, with imperfect thermostatic conditions we acknowledge that we cannot 100% guarantee the Birch effect was the sole factor influencing soil respiration. We propose to reword instances where the Birch effect is discussed **more conditionally** in the **abstract**:

L24-30: One possible explanation could be that rewetting may have triggered a stronger mineralization response, commonly known as the Birch effect, in drier topsoils compared to conditions where the soil remained consistently wetter with a shallower GWT level. Based on our findings, including the process of texture-specific capillary supply from the GWT is required to adequately simulate moisture in the topsoil during droughts as they occurred over the past summers in North-West Europe, depending on GWT and texture combination. However, the net effect on topsoil carbon mineralization is complex and warrants further investigation, including the integration of processes related to fluctuating soil moisture following rewetting.

, in the **discussion**:

L493-500: Accordingly, a potential explanation could be that the drier condition of topsoil with deeper GWT may have amplified the Birch effect, i.e. rewetting C mineralization pulses caused by the watering events. In the light of this, it is possible that the expected enhancement of C mineralization (as represented by $k_s$) due to increased moisture from capillary action under shallower GWT depths was offset in loamy sand and sandy loam soils, and perhaps even overruled in silt loam soils, by a stronger Birch effect in drier soils with less capillary moisture supply. In other words, the reduced mineralization rates observed in silt loam under a shallower GWT ($k_s$) might be explained, at least in part, by a larger capillary moisture supply compared to the other soil texture and GWT combinations, although other factors, such as an intertwined effects between moisture and temperature, could also be at play.

and in the **conclusion**:

L543-548: Moreover, we found that after rewetting, C mineralization pulses were larger for the deepest GWT treatment. We hypothesize that, as soil was drier for the deeper GWT owing to less capillary moisture supply, the imposed rewetting events caused stronger Birch effects. Hence, not just topsoil moisture state itself, but also the extent of its fluctuation over time, probably culminates into the overall observed net effect of GWT depth onto C mineralization. To use empirical data from experiments like this to improve soil models, further studies will be needed to deconvolute the effects of soil moisture regimes on soil C mineralization.

*246 Are measured (saturated) hydraulic conductivities available? In combination with the measured soil water retention functions this would allow to estimate the infiltration depth, or even a soil water content profile over depth. To get an idea of how much the upper 20 cm with the labeled C were actually affected by the irrigation for the different soils…*

*315 well, now you end up with two factors affecting (or not affecting) the water content in the upper 20 cm with labelled C. The ground water table AND the irrigation. The irrigation will affect the respiration of the different soils differently. This might be difficult to interpret in the end.*

*500 but not only the GWTs were modified, simultaneously precipitation was simulated*

➔ We here below address these three comments by referee 2 together, as they are closely related. We firstly want to clarify that the exact same amount of water was irrigated to each of the three soil textures, moreover following the same timing for both GWT batches. Hence irrigation dose and timing was kept identical across the GWT x texture treatments and should not be a factor in our experiment. To **evaluate** the impact of these **wetting events on soil moisture for the three differently textured soils**, we now conducted extra **simulations** using **HYDRUS-1D**. Because saturated hydraulic conductivities were not measured for our topsoil layer we estimated them using the Rosetta pedo-transfer function (Schaap et al., 2001), based on artificial neural network analysis, using texture data, bulk densities and water contents at field capacity and wilting point. We limited these simulations to the top 30 cm of the soil profiles, as the moisture sensors indicated that irrigation impacts would be negligible below this depth (i.e. there was no further fluctuation in the VWC at 30 cm after adding water). Simulated moisture content and hydraulic head along the soil profiles at various time points (0.5, 2, 6, 12, 24, 36 hours) after adding water (at time 0 h) are shown in the figures below. According to these simulations half an hour after water was added (blue line), it reached a depth of 15 cm in loamy sand, 11 cm in sandy loam, and 7 cm in silt loam soils. Over time, the water infiltrates to greater depths. Ryegrass-C mineralization measurements were taken no earlier than 24 hours after wetting, when the added water had ample time to infiltrate up to a depth of 20 cm for all three soil textures. In the loamy sand and sandy loam soils, by 24 hours the water was fairly evenly distributed within the 0-20 cm depth. In silt loam, a more uniform distribution was achieved by 36 hours after wetting. We propose to **add this figure of the simulations to the appendix**, and **add** the following six lines **to** the **M&M** section:

L150-155: Moisture measurements at −10 cm depth were considered representative for the entire repacked topsoil layer (0 to −20 cm), and therefore suitable for assessing [13]C-labeled ryegrass mineralization within this layer. Simulations in HYDRUS-1D confirmed that moisture had infiltrated to about 20 − 30 cm and the water content had become nearly constant with depth across the 0 − 20 cm layer 24 − 36 h after water addition (Fig. A1). Since soil $CO_2$ efflux

[Figure]

measurements (see 2.2.2) were taken then to the earliest after watering events, the −10 cm measurements provided a representative state of the topsoil moisture state.

Figure A1: Simulations using HYDRUS-1D to determine when the water, added at T0, reached the lower boundary of the repacked topsoil containing $^{13}$C-labeled ryegrass (at −20 cm depth). A 24-hour period after water addition was found sufficient for all three soil textures (loamy sand, sandy loam, and silt loam) to reach an approximately constant moisture level across the topsoil layer.

*406 No, that is not unlikely. This could be a result of macropores.*

➔ We recognize that macropores can serve as preferential flow paths, facilitating a downward water flux through the soil pore network. However, in combination with the volumetric soil moisture data, we think it could not have been a dominant avenue for moisture transport. Higher measured water contents for the –115 cm GWT treatment compared to the –165 cm GWT treatment demonstrate that the net transport of moisture must have predominantly been upwards. **We propose to include this remark the following way in the discussion text**:

L423-429 These observations imply a downward moisture transport near the GWT, which could be the case if macropores served as preferential flow pathways. However, as we observed an increase in water content during the shallower GWT treatments above this soil segment, upward moisture movement would have been the dominant water flow, with the GWT as starting point of the flow path. We believe these unexpected negative ΔH values are therefore more likely a result of cumulative errors when converting measured $\theta_V$ into H using soil water retention curves obtained from drying phases. Due to hysteresis, such curves can be inaccurate for soil wetting (Hillel, 2003), which was the main expected process at the considered deeper depths near the GWT.

*409 well, there are very small lab tensiometers on the market.*

➔ We **propose to add four lines to the discussion text** regarding this matter:

L429-433: In retrospect, a better experimental approach would have been to directly measure hydraulic head using compact tensiometers, especially in the undisturbed parts of the soil columns, where following moisture transport was the main objective. However, even for the smallest lab-scale models, their installation would likely have caused significant disturbance to the soil columns, in contrast to the volumetric sensors used in this study, which had very sharp fine rods able to pierce the waterproof foil surrounding the soil columns with minimal disruption.

*435 these 'bell-shaped' curves were usually determined for disturbed soil samples. And in the wet range the decrease of respiration is rather related to an oxygen deficit and is actually not related to water content. For undisturbed samples the moisture sensitivity function of respiration actually looks different.*

➔ We believe we have clearly stated that the repacked topsoil layer from which we monitored ryegrass-C mineralization was in fact 'disturbed' , as already explained in L124-125 and L135-136 in the M&M section. We did not further modify the revised manuscript accordingly.

**RC3**:

*L121 I was disappointed that the authors did not report their results for the mineralization of native SOC and only show the $^{13}CO_2$ results. This would have given a fuller picture of C dynamics in their systems. Also, because the added $^{13}C$ litter is likely to 'exist' as particulate organic matter and a large proportion of SOC 'exists' as mineral associated organic matter, quantifying their relative mineralization under different GWT treatments could have greatly improved your hypotheses and results, especially given the lack of difference in $^{13}C$ mineralization between soils. Since that data surely exists and does not require repeating or doing new experiments, I highly encourage the authors to include this data. The authors` concerns regarding soil properties effects on SOC mineralization can be partially addressed by normalizing CO2 to C content, and besides the mineralization of the 13C litter is likely also impacted by various properties.*

➔ $CO_2$ efflux from native SOC was higher during the first GWT treatment batch (deeper GWT with lower topsoil $\theta_V$) compared to the second one (shallower GWT with higher $\theta_V$) for all three textures. This suggests that a substantial portion of the native SOC derived $CO_2$ efflux resulted from SOC from the soil below 20 cm, which had not been renewed between both GWT-treatments. If we would have intended to as suggested by the referee compare native SOC mineralization, we should have used completely fresh soil columns for both GWT-treatments. Aside from considerable practical constraints (collecting 0-2 m intact soil cores is not trivial) we prioritized assessing the effect of GWT depth on identical columns to eliminate the potential impact of soil heterogeneity on moisture dynamics. This setup trade-off was initially decided, and is also the very reason why we worked for each batch with new $^{13}C$-labeled ryegrass and topsoil material. For clarification we propose to **add the following** 3 lines to the M&M:

L130-135: We used a model OM substrate (in casu $^{13}C$-labeled clipped ryegrass) and did not compare native soil OM mineralization, as its inherently different quality and quantity between the three soils would no longer allow studying the effect of GWT depth, soil texture and their interaction on soil OM. The native soil OM-derived $CO_2$ efflux in part originated from mineralization of native soil OM in the undisturbed (−20 to −200 cm) soil. As this soil column was not renewed between both GWT treatment batches, the quality of subsoil OM differed between both GWT treatments. For these reasons it is not meaningful to present native SOC mineralization data and results are kept restricted to ryegrass C-mineralization.

*L180 delta13C of CO2 undergoes fractionation at different diffusivities (e.g. water contents). https://agupubs.onlinelibrary.wiley.com/doi/full/10.1029/2008JG000766*

*Given that your labeled material wasn't very highly enriched, such differences can have an effect on your calculations. Were the parallel incubations of end members (L179-180) done at the same GWTs as the experimental incubations?*

➔ We assessed the isotopic signature of $CO_2$ emitted from native SOC and ryegrass using parallel identical 20 cm packed soil columns, so not in the 2 m soil columns, as described in L180-186. Working with soil on top of undisturbed 180 cm soil columns in GWT barrels, would have been complicated as there would also be 'contamination' of $CO_2$ being emitted from the underlying soil, limiting to precisely assess the $\delta^{13}C$ of $CO_2$ derived from ryegrass mineralization. Nevertheless, in these ancillary soil core incubations care was taken to as closely as possible simulate the moisture regime as it occurred in the main experiment: viz. every few days the 20 cm soil columns were weighed, and (a small amount of) water was added if their $\theta_v$ became lower than certain thresholds (0.1, 0.12 and 0.15 $m^3$ $m^{-3}$ for respectively loamy sand, sandy loam and silt loam soils), while also the larger wetting events were applied identically as in the 2 m columns. To now better clarify this, we propose to **add the following** 2 lines to the M&M:

L195-200: The isotopic signature of $CO_2$ emitted from either end member, i.e. $\delta^{13}C\cdot CO_{2\cdot ryegrass}$ and $\delta^{13}C\cdot CO_{2\cdot SOC}$, were analyzed in ancillary soil incubations as in Mendoza et al. (2022). The $\delta^{13}C\cdot CO_{2\cdot SOC}$ was determined in parallel 20 cm packed soil columns with no ryegrass added. For $\delta^{13}C\cdot CO_{2\cdot ryegrass}$, such soil columns were amended with a high dose of ryegrass (3 g C $kg^{-1}$ soil; indicated as "high"). The rationale for using smaller columns compared to the main experimental setup was to exclude $CO_2$ emissions from the underlying soil OC, ensuring a more accurate assessment of $\delta^{13}C\cdot CO_{2\cdot ryegrass}$.

---

## Author Response (AR3)

We would like to thank the editor for the valuable suggestions, which we have thoughtfully incorporated. Below, we present the editor's comments in *italics*, followed by our responses and proposed revisions to the text:

*I think that you replied to the comments of the reviewers but I propose that you revise your response to the question of reviewer 3 with respect to the monitoring of native C mineralization. I think that also the drastic change in temperature from field conditions to a uniform temperature of 20 °C over the entire profile length is important. This change in temperature will especially have an impact on the carbon mineralization in the deeper soil horizons.*

*Ln 145: 'The native soil OM-derived CO2 efflux in part originated from mineralization of native soil OM in the undisturbed (–20 to – 200 cm) soil. As this soil column was not renewed between both GWT treatment batches, the quality of subsoil OM differed between both GWT treatments. For these reasons it is not meaningful to present native SOC mineralization data and results are kept restricted to ryegrass-C mineralization.' I am not so convinced by this reply to the reviewers' question why you did not consider the native CO2 efflux. Isn't the C-pool large enough so that no big effect in C pool size and composition should be expected after a few weeks? I would however expect that the change in temperature of the subsoil from roughly 10 °C to 20°C would impact the mineralization of subsoil carbon a lot and would rather consider this as an argument why not to look at the native carbon mineralization.*

We agree that a uniform temperature of 20°C throughout the entire soil profile is unrealistic and can see that this forms a valid argument to disregard SOC emissions. We do still believe that even at lower realistic subsoil temperature then still our serial experimental design would not have allowed comparing SOC-derived mineralization between both GWT-batches. Although indeed the background SOC stock would not be different between the batches, there will be differences between the labile-C pool that is likely to have been in part depleted after the first batch. We therefore also still included our other motivation in the M&M text.

➔ To clarify, we propose the following revisions to L130-142:

'We used a model OM substrate (*in casu* [13]C-labeled clipped ryegrass) and focused solely on the mineralization of this added substrate in the topsoil, without including data on native soil OM mineralization. This approach was chosen primarily because the inherently different quality and quantity between the three soils would no longer allow studying the effect of GWT depth, soil texture and their interaction on soil OM mineralization. Additionally, $CO_2$ effluxes from native soil OM included contributions  from mineralization in the undisturbed subsoil column (from –20 to – 200 cm), which was left unchanged between GWT treatment batches to reduce potential soil heterogeneity influences on moisture dynamics.  As native soil OM in this subsoil had thus already undergone partial mineralization during the first GWT treatment, starting conditions would differ for the second GWT treatment. Particularly so as subsoil was moreover kept uniform at 20 °C, which deviates strongly from expected field conditions, where temperature is lower and typically decreases with depth. This discrepancy further limits the applicability of our setup for assessing subsoil OM mineralization.

*I propose also to reconsider the name of the model that you use: parallel first-zero-order kinetic model. In fact, since you express the mineralized carbon as a percentage of the applied carbon, you assume that the mineralization scales linearly with the initial carbon amount. This would correspond with a first order kinetic. In a zero-order kinetic, the reaction rate is not dependent on the concentration of the substrate. In your model it is proportional to the substrate concentration. But, what you consider as a 'zero-order' is in fact a linearization of a first-order reaction for short times compared to the reaction time scale. Therefore, I propose calling the model a parallel two rate first-order kinetic model.*

➔ We agree and adapted the phrasing in L214, L246, L248, L374, L384 and L386.

*Finally, I think you need to define better what you mean by 'capillary rise'. It could be interpreted as an upward water flow or it could be the hydrostatic water content profile above a groundwater table. When you assume hydrostatic conditions, the water content at the soil surface can be read for a given groundwater table depth from the water retention curve. But, when there is this upward flow and evaporation, the water content at the soil surface will always be smaller than the water content for the same groundwater table depth under hydrostatic conditions. In order to assess the water content under upward flow conditions near the soil surface, it is important to know the unsaturated hydraulic conductivity and the evaporation flux. I propose to include information about the upward evaporation fluxes that were measured in the different treatments. It should be possible to derive that from the amount of water that was added at the bottom of the columns to keep the water table constant over time.*

➔ In this study, we consider capillary rise as the active upward movement of water driven by capillary forces. This process must indeed be considered alongside evaporation, as some of the moisture supplied by capillary rise is subsequently lost to evaporation. This is also described at the end of the introduction in one of our hypothesis in L 90-91:

'We expected an increasing susceptibility to reduced moisture of the C mineralization with coarser soil texture as water losses by evaporation would be less compensated by capillary moisture input.'

In the discussion, specifically in section 4.1, lines 407-409, we also distinguish between the potential capillary rise driven by hydraulic head differences, which depend on the hydraulic characteristics of the soil, and the actual net effect on topsoil moisture due to to evaporative losses:

'First, although positive hydraulic head differences (ΔH) between −60 cm and −30 cm enabled capillary action, they displayed an increasing trend throughout the experiment (Fig. 3). Hence, the soil was observed to be drying out at 105 cm and 135 cm above the GWT, indicating clearly that evaporative losses were insufficiently compensated by a capillary water flux. '

**However, to make this clear this from the beginning of the manuscript**, we propose **modifying** the term 'capillary rise' in the following lines of the **abstract and introduction**:

L15: We examined (1)  upward moisture flow by capillary  action along the soil profile and specifically into the top 20 cm soil, and (2) consequently the effect of GWT on decomposition of an added $^{13}$C-enriched substrate (ryegrass) over a period of ten weeks, with limited wetting events representing a dry summer.

L23: These findings suggest that the  upward capillary moisture flow, along with the resulting increase in topsoil moisture and the anticipated enhancement of biological activity and ryegrass mineralization, might have been counteracted by other processes.

L 45: Whether or not moisture supply via upward capillary  flow is a relevant process to be accounted for by soil C models, will not only depend on climate, but also on factors such as the depth of the GWT and soil physical properties.

L53: To the best of our knowledge, there exists no robust proof on whether or not, and when, GWT depth might significantly control topsoil heterotrophic activity. Such insights are essential to determine whether incorporating GWT depth and upward capillary  moisture flow in updated soil C models is warranted.

L67: This limitation restricts our ability to study the effect of individual components of the soil water balance like capillary  moisture supply.

L76: In sum, there is little empirical evidence of the control of moisture dynamics by capillary  water flow on topsoil organic matter (OM) mineralization. Not only the impact of GWT onto mean topsoil water content seems a blind spot, but possibly also the amplitude of soil moisture fluctuation in topsoil may depend on the magnitude of moisture supply by capillary  action.

L84: Our main aim was to study if, during a (simulated) period with limited rainfall, there would be a significant effect of capillary  moisture flow from the GWT on topsoil moisture and OM mineralization for loess deposited arable lands in North-West Europe.

And propose the following adaptations in the **discussion and conclusion**:

L418: When the GWT was raised to −115 cm, moisture at −30 cm was higher than at the −165 cm GWT, implying that upward capillary moisture flow  markedly impacted soil moisture up to at least 85 cm above the GWT, less so beyond 105 cm and no more beyond 135 cm.

L422: Hence, it seems likely that upward moisture  flow in the sandy loam and silt loam columns by capillary  action reached at least up to a height of 135 cm.

L458: Consequently, as we worked with 2 m undisturbed soil columns and realistic GWT depths, we do expect findings of GWT-dependent capillary moisture supply  and topsoil moisture to be representative for the field situation.

L517: According to our results, hydrological modules which calculate water fluxes between adjacent soil layers, but with free-draining lower boundaries, applied in some soil C models, e.g. DAYCENT (Schimel et al., 2001), would be less accurate for simulation of topsoil moisture during periods with limited rainfall, as these models do not incorporate capillary moisture flow  in simulating recharge and presuppose that water draining from the soil profile is lost.

L548: For situations where the GWT is within these ranges our findings should motivate to include bidirectional water flow, i.e. drainage and capillary transport , in soil models.

L555: During prolonged periodic droughts, expected to become more frequent under future climate, correct simulation of the mostly neglected capillary  moisture  transport may become imperative for reliable simulation of C cycling in agricultural land in North-West Europe.

**To get back to your proposal to indirectly quantify evaporation**: it would indeed have been very informative to establish a water balance over time, including an indirect **estimation of soil evaporation** by measuring the amount of water added to the barrels to maintain a constant level. However, because the barrels had a much larger diameter than the four soil columns per texture positioned within each barrel (Fig. 1), a considerable amount of water evaporated directly from the open water surface inside the barrels, rather than solely from evaporation out of the soil columns. As a consequence, our setup does not allow deriving evaporation based on water level changes in the barrels.

*Detailed comments*

*Ln 22: 'In contrast, C mineralization pulses after the wetting events were even higher for the drier −165 cm GWT soils. For the silt loam soil, where capillary rise supply had the largest contribution to topsoil moisture, a lower mineralization rate of the stable Cryegrass pool was also found with shallower GWT.' These sentences are not clear. First you write that the mineralization pulses after wetting are higher for the -165 cm soils. Then you write that in the silt loam soil, the mineralization is lower for the shallower GWT. Do you mean that the C mineralization pulses of the labile pools after wetting events are higher for lower groundwater tables in all soils and that for the silty-loam soil also the mineralization of the stable pool is higher for the deeper GWT than for the shallower GWT? Or do you mean that for the silty loam soil the mineralization of the stable pool is lower than in the other soils for the shallower GWT? Make clear what you are comparing with each other when you write larger than or smaller than ... .*

➔ We propose to adapt our text the following way:

L21-L24: 'In contrast, $CO_2$ efflux pulses after some of the wetting events were even higher for the drier −165 cm GWT than for the −115 cm GWT across all three soil textures. Additionally, a model fitted to cumulative ryegrass mineralization showed a lower mineralization rate for the stable $C_{ryegrass}$ pool in the silt loam soil with the shallowest GWT, where capillary rise contributed most significantly to topsoil moisture, compared to other combinations of soil texture and GWT depth.'

*Ln 23: 'These findings suggest that a potential capillary rise effect of increased topsoil moisture on ryegrass mineralization might have been counteracted by other processes.' You need to mention which effect has been counteracted because this is not clear. I propose: 'These findings suggest that a potential capillary rise effect of increased topsoil moisture that leads to higher biological activity and ryegrass mineralization have been counteracted by other processes.'*

➔ We propose to adapt the text the following way:
L24: 'These findings suggest that the upward capillary moisture flow, along with the resulting increase in topsoil moisture and the anticipated enhancement of biological activity and ryegrass mineralization, might have been counteracted by other processes.'

*Ln 144: shouldn't you keep 'mineralization'?*

➔ We agree and adapted as suggested.

*Ln 157: I suppose the dose was 1.5 g C kg-1 of dry soil (and not wet soil).*

➔ Correct, we have clarified this in the text.

*Ln 155: 'texture-specific soil-grass mixtures' could also be interpreted as the soil-grass mixtures being different for the different textures, which was not the case. I propose: the soil-grass mixtures for the three soil textures.*

➔ We agree and adapted as suggested.

*Ln 242: 'hydraulic head differences (ΔH) between two adjacent sensor positions above the GWT' This is not clear because you can calculate differences between two adjacent sensors in two opposite ways. Since you are interested in the sign of the difference, it is important to mention how you calculate the difference. A way to avoid this is to mention that you calculated averaged hydraulic gradients between two measurement heights, where height was defined to increase in the upward direction (as opposed to depth which increases in the downward direction).*

➔ We agree and adapted as suggested.